# LRAgent: Efficient KV Cache Sharing for Multi-LoRA LLM Agents

Hyesung Jeon [1]   Hyeongju Ha [1]   Jae-Joon Kim [1]

## Abstract

Role specialization in multi-LLM agent systems is often realized via multi-LoRA, where agents share a pretrained backbone and differ only by lightweight adapters. Despite sharing base model weights, each agent independently builds and stores its own KV cache for the same long, tool-augmented trajectories, incurring substantial memory and compute overhead. Existing KV cache sharing methods largely overlook this multi-LoRA setting. We observe that, cache differences across agents are dominated by adapter outputs, while activations from the shared pretrained backbone remain highly similar. Based on this observation, we propose LRAgent, a KV cache sharing framework for multi-LoRA agents. It decomposes the cache into two components, a shared base component derived from pretrained weights and an adapter-dependent component derived from LoRA weights. LRAgent reduces memory overhead by sharing the base component across agents and storing the adapter component in its inherent low-rank form. It also reduces computational overhead by sharing the low-rank cache, enabled by a shared-A multi-LoRA architecture. This avoids redundant computations for contexts that have already been processed by other agents. To efficiently reconstruct adapter contributions at runtime, we introduce Flash-LoRA-Attention, a kernel that reorders attention computation to avoid materializing the low-rank cache to full dimension. LRAgent achieves throughput and time-to-first-token latency close to fully shared caching, while preserving accuracy near the non-shared caching baseline across agentic question-answering benchmarks. Code is available in the official repository.

## 1. Introduction

Recently, LLMs have been widely adopted in agent systems due to their long-context understanding ability (Dubey et al., 2024; Jiang et al., 2023; Yang et al., 2025a), reasoning ability (Wei et al., 2022; Yao et al., 2023a; Snell et al., 2024), and external tool interaction capabilities (Yao et al., 2023b; Shen et al., 2024; Qin et al., 2024). In particular, multi-LLM agent systems have gained increasing attention for their ability to assign specialized roles to multiple agents that collaboratively decompose and solve complex tasks (Talebi-rad & Nadiri, 2023; Wu et al., 2024; Rasal, 2024; Zhang et al., 2025). These agents retrieve information from external tools, augment it with generated outputs, and pass the accumulated context, referred to as trajectories, to other agents for subsequent steps. To improve accuracy, a common approach is to fine-tune a pretrained model separately for each agent role. These fine-tuned models are typically trained on pre-generated trajectories that reflect role-specific behavior and tool usage patterns (Shinn et al., 2023; Bo et al., 2024; Liu et al., 2025a; Bai et al., 2025; Fu et al., 2025a). Parameter-efficient fine-tuning (PEFT) methods, such as low-rank adaptation (LoRA) (Hu et al., 2022), further enhance scalability by reducing the number of trainable parameters from the full model to a pair of low-rank matrices. As a result, multi-LoRA architectures enable agents to share the large pretrained backbone during inference while retraining lightweight, role-specific adapters (Wang et al., 2023; Xia et al., 2024). This design has proven effective in practice, consistently outperforming single-model agents and non-fine-tuned baselines in agentic tasks (Qiao et al., 2024; Yu et al., 2024; Liu et al., 2025b; Li et al., 2025).

Due to the long trajectories in LLM agent systems, KV cache overhead and compute overhead become more severe in multi-agent systems than in single-agent settings, because each agent maintains its own KV cache and redundant prefills occur even though a large portion of the context is shared. This redundancy increases both memory usage and inference latency. To mitigate the memory issue, recent work has explored KV cache sharing across agents. However, existing approaches either require architectural modifications and additional training for cache fusion (Woo et al., 2025; Fu et al., 2025b), focus mainly on handling positional misalignment caused by agent-specific prefixes (Yang et al., 2025b; Pan et al., 2025; Ye et al., 2025), or rely on selective

[1]Department of Electrical and Computer Engineering, Seoul National University, Seoul, Korea. Correspondence to: Hyesung Jeon <hjeon2k@snu.ac.kr>, Hyeongju Ha <mnv1009@snu.ac.kr>, Jae-Joon Kim <kimjaejoon@snu.ac.kr>.

*Proceedings of the $43^{rd}$ International Conference on Machine Learning*, Seoul, South Korea. PMLR 306, 2026. Copyright 2026 by the author(s).

recomputation of certain tokens or layers (Yao et al., 2025; Liu et al., 2026). Furthermore, these works focus primarily on memory reduction but still incur redundant computation to build a hidden state for a context that has already been processed by other agents. Importantly, KV cache sharing schemes that explicitly exploit the multi-LoRA architecture remain largely unexplored.

In this work, we make a key observation that, for the same context, cache discrepancies across agents are dominated by task-specific LoRA-induced outputs, while activations produced by the shared pretrained backbone remain highly similar. Motivated by this observation, we propose LRAGENT, a KV cache sharing framework tailored to multi-LoRA agent systems. We decompose the cache into two parts: a shared component computed from the pretrained weights, which we call the base cache, and an agent-dependent component induced by the LoRA weights, which we call the adapter outputs. The next key property we exploit is that the adapter output naturally admits a low-rank representation. Specifically, we store the intermediate activations produced right after the LoRA down-projection, which have a small rank dimension. We refer to these activations as the LR cache. At runtime, we reconstruct the full-dimension adapter contribution from LR cache by multiplying it with the LoRA up-projection matrix only when needed. As a result, we compress multiple KV caches into a single shared base cache with lightweight LR caches. Based on this concept, we introduce two cache sharing schemes. *BaseShared* shares the base cache across all agents while maintaining a separate LR cache per agent, substantially reducing KV cache memory. Furthermore, motivated by recent multi-LoRA variants that share the down-projection matrix across tasks (Tian et al., 2024; Yang et al., 2025c), we extend our idea to *BaseLR-Shared*, which also shares the LR cache as well as the base cache by aligning agents to use a common down-projection. This extension further reduces both memory usage and the amount of computation for previously seen contexts. To minimize the runtime overhead of reconstructing adapter contribution using LR cache, we design Flash-LoRA-Attention, which reorders attention computation to avoid materializing low-rank caches to full dimension and implements this strategy efficiently on top of FlashAttention (Dao et al., 2022; 2023). Overall, our approach enables KV cache sharing tailored to the multi-LoRA architecture, achieving memory and inference efficiency close to fully shared KV caching while preserving accuracy near the non-shared KV baseline.

## 2. Background

### 2.1. Multi-LoRA Architecture

**LoRA** LoRA (Hu et al., 2022) is a PEFT method that adapts model weights by adding a pair of low-rank matrices to the frozen pretrained base weights. Formally, LoRA parameterizes the weight update as:

$$W = W_0 + \Delta W, \quad \Delta W = AB, \tag{1}$$

where $W_0 \in \mathbb{R}^{d_{\text{in}} \times d_{\text{out}}}$ is the pretrained base weight, $A \in \mathbb{R}^{d_{\text{in}} \times r}$ is the down-projection matrix, and $B \in \mathbb{R}^{r \times d_{\text{out}}}$ is the up-projection matrix, with rank $r \ll \min(d_{\text{in}}, d_{\text{out}})$. Since only $A$ and $B$ are optimized instead of $W_0$, it significantly reduces the number of trainable parameters, leading to lower memory usage and computation compared to full fine-tuning. In particular, applying LoRA to the query and value projections yields the best accuracy for a given number of parameters and is therefore widely used in practice. Unless otherwise stated, we apply LoRA to the query and value projections in all experiments.

**Multi-LoRA** A typical multi-LoRA system augments a pretrained base weight with multiple task-specific low-rank weights (Xia et al., 2024). For each task index, or agent role index in our setting, $i \in \{0, 1, \dots, N-1\}$, where $N$ is the number of tasks, and given LoRA weights $A_i \in \mathbb{R}^{d_{\text{in}} \times r}$ and $B_i \in \mathbb{R}^{r \times d_{\text{out}}}$, the task-specific weight used for inference is:

$$W_i = W_0 + \Delta W_i, \quad \Delta W_i = A_i B_i. \tag{2}$$

Given an input activation tensor for task $i$, $X_i \in \mathbb{R}^{l \times d_{\text{in}}}$, with sequence length $l$, the output $Y_i \in \mathbb{R}^{l \times d_{\text{out}}}$ and the adapter output $\Delta Y_i$ are computed as:

$$
\begin{aligned}
Y_i &= X_i W_i = X_i W_0 + (X_i A_i) B_i, \\
\Delta Y_i &= X_i \Delta W_i = (X_i A_i) B_i,
\end{aligned}
\tag{3}
$$

where $X_i A_i \in \mathbb{R}^{l \times r}$ is the intermediate activation produced by the down-projection.

**Multi-LoRA with Shared-$A$** Recent works report that task-specific differences in multi-LoRA systems are primarily driven by the up-projection matrices $B_i$, while the down-projection matrices $A_i$ encode highly similar intrinsic information. Hence, sharing a down-projection matrix $A$ can improve accuracy compared to conventional multi-LoRA systems, as $A$ becomes more generalizable across tasks. This effect is not limited to specific datasets or subtasks, but is effective across diverse domains and datasets (Tian et al., 2024; Yang et al., 2025c). We therefore adopt the shared-$A$ design for improved accuracy in our setting, while simplifying it to match the conventional multi-LoRA architecture. Specifically, we replace the token-wise LoRA routing used in prior work with a sequence-wise assignment, since agent roles are predefined and do not require token-dependent routing. This yields the same overall architecture as conventional multi-LoRA, except that the down-projections share weights. We design cache sharing strategies that support both standard multi-LoRA and the shared-$A$ variant, with the latter further leveraging the efficiency benefits of our approach. Overall, our method reduces memory usage and latency while preserving each agent's role-specific behavior.

## 2.2. Multi-LLM Agent KV Cache Sharing

**Multi-LLM Agent** Multi-LLM agent systems can be implemented either with a single model that plays different roles via role-specific system prompts, which can be viewed as a form of in-context learning, or with multiple models, often by fine-tuning the same backbone model for different roles (Talebirad & Nadiri, 2023; Wu et al., 2024; Shinn et al., 2023; Liu et al., 2025a; Fu et al., 2025a). These systems commonly use three highly heterogeneous and irreplaceable agent roles: a planning agent for reasoning, an action agent for tool use, and a reflection agent for revising the answer. Methods that incorporate fine-tuning for role specialization often synthesize agent-trajectory datasets using instruction-following models (Touvron et al., 2023; OpenAI et al., 2024; DeepSeek-AI et al., 2024), and then fine-tune with PEFT methods such as LoRA. These approaches have been shown to outperform both single-model agents and non-fine-tuned baselines (Qiao et al., 2024; Yu et al., 2024; Liu et al., 2025b), leading to broad adoption in practice. In this work, we follow AutoAct (Qiao et al., 2024) and fine-tune LoRA adapters for the plan, action, and reflection agents using the provided agent trajectory dataset.

**KV Cache Sharing** Meanwhile, due to long trajectories from multi-step reasoning and multiple retrieval of large contexts from external tools, memory and compute overhead becomes much more pronounced in multi-agent settings than in single-agent scenarios. This is because each agent maintains its own KV cache even though much of the context is shared. Moreover, the same context is processed independently by multiple agents, leading to substantial computation. Together, these effects introduce memory and compute redundancy, increasing memory usage and latency. To mitigate the memory issue, recent works have explored KV cache sharing for context that is shared across agents. Most approaches either introduce new model architectures that require additional training to enable cache sharing, or are limited to handling positional misalignment so that precomputed KV caches for overlapping context chunks can be reused within a single model. ICaRus (Woo et al., 2025) proposes a decoder architecture that fine-tunes only the query projections for downstream tasks, enabling agents to share an encoder-generated cache and reconstruct task-specific additive caches at runtime. Cache2Cache (Fu et al., 2025b) adds a trainable linear layer that project and fuse a source model's KV cache into a target model. KVLink (Yang et al., 2025b), KVFlow (Pan et al., 2025), and KVComm (Ye et al., 2025) address positional misalignment from agent-specific prefixes by aligning cache offsets and adjusting positional embedding at runtime, allowing reuse of overlapping context despite divergent prefixes within a single model. We note that these prefix-aware methods rely on differences in input prefixes, and therefore reduce to a naive fully shared KV cache baseline when agents share identical

*Table 1.* Average cosine similarity across agent pairs for the base cache, full cache, and adapter output, on the same context.

| Model | Full cache | Base cache | Adapter output |
|---|---|---|---|
| LLaMA-3.1-8B-Instruct | 0.9576 | 0.9726 | 0.0538 |
| Ministral-8B-Instruct | 0.9200 | 0.9530 | 0.0225 |

system prompts. Other works rely on selective recomputation. For instance, CacheBlend (Yao et al., 2025) reuses KV caches by recomputing a small subset of tokens that are critical for accuracy. DroidSpeak (Liu et al., 2026) enables KV cache reuse across fine-tuned LLMs that share the same backbone by selectively recomputing KV cache of a predefined set of critical layers while reusing the KV cache for the remaining layers, reporting higher accuracy than token-wise recomputation methods such as CacheBlend. In addition, DroidSpeak introduces a hidden state cache to skip recomputation for the initial few non-recomputed layers and directly feed to the first recomputed layer. However, although DroidSpeak is the closest to our work, it still processes hidden states for already seen context, similar to other approaches, so it reduces only the key and value projections for cache updates while leaving most computation unchanged. This highlights that fully reusing KV caches to eliminate computation for redundant context across models remains challenging. Moreover, KV cache sharing tailored to multi-LoRA systems remains largely overlooked despite their wide deployment in practice. To the best of our knowledge, this is the first work that explicitly tailors KV cache sharing to multi-LoRA agent settings and our work is complementary to prior cache-sharing methods.

## 3. Methodology

### 3.1. Cache Similarity Across the Agents

We first demonstrate that, for the same context, differences in cache values mainly arise from the adapter output, which is small in magnitude (see Appendix A.1) but contains critical agent-specific information. Applying Equation 3 to the value projections, we let $X_i$ denote the hidden states obtained by running agent $i$, which exhibits high similarity across agents (see Appendix A.2). In this setting, Figure 1 and Table 1 show that the base cache $Y_{\text{base},i} = X_i W_0$ remains highly similar across agents on the same context. In contrast, the adapter output $\Delta Y_i = X_i \Delta W_i$ acts as a largely decorrelated perturbation, with cosine similarity close to zero. With these small, decorrelated adapter output, the expected cosine similarity of the full cache $Y_i = Y_{\text{base},i} + \Delta Y_i$ is lower than that of the base cache, empirically by about 3% on average, where concrete derivation is provided in Appendix A.3. This motivates sharing only the base cache, rather than the full cache, to better preserve the small but critical agent-specific contributions from the adapters. Otherwise, the discrepancy accumulates over iterative agent

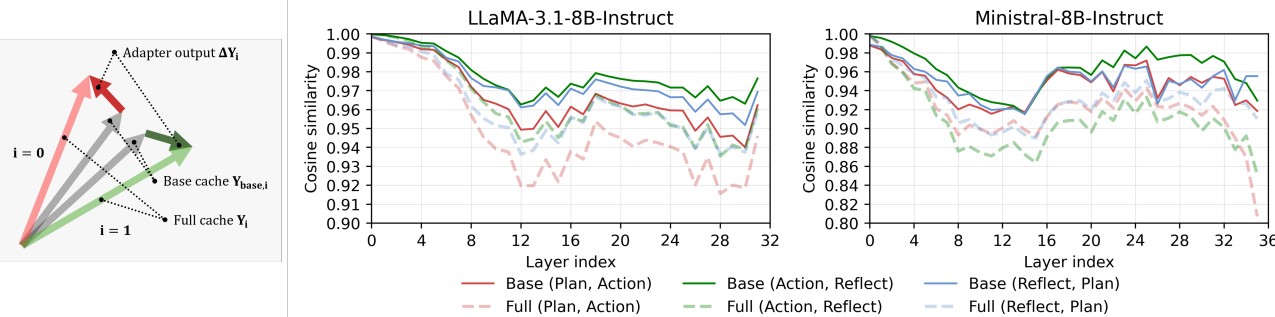

*Figure 1.* **(Left)** Relationship between the full cache, base cache, and adapter output. **(Right)** Layer-wise pairwise cosine similarity of the base and full caches, measured on the same context across three agent pairs using 128 samples of 2k tokens from the HotpotQA dataset.

executions and can lead to a noticeable accuracy drop. We also note that the cosine similarity of the key cache remains above 0.98 on average (see Appendix A.4), suggesting that value cache management is the key factor for preserving accuracy in multi-agent inference. we observe a similar pattern in other agent role configurations (see Appendix A.5).

### 3.2. BaseShared and BaseLRShared

In this section, we present KV cache sharing schemes tailored to the multi-LoRA architecture. Our key idea is to decouple the value cache into a shared component (base cache) produced by the pretrained weights, and an adapter-dependent component. We reuse the base cache across agents without recomputation, and store the adapter-dependent component in a compact low-rank form (LR cache) that is expanded to full-dimension form on demand at runtime. We first introduce `BaseShared`, which primarily reduces memory usage, and then extend it to `BaseLRShared`, which leverages shared-$A$ multi-LoRA architecture and further reduces both computation and memory usage. Figure 2 illustrates the agent execution flow and the corresponding cache management in our methods, which we discuss in detail in the following paragraphs.

**BaseShared** We first decouple the value cache into a base component $Y_{\text{base}}$ and an adapter-dependent component $\Delta Y_i$. Here, we share a single base cache across agents even though the layer inputs $X_i$ are not exactly identical, motivated by the observations in Section 3.1, which show that the base cache $X_i W_0$ remains highly similar across agents on the same context and that the remaining differences are dominated by the adapter contribution. Concretely, for each newly appended context, the agent that processes the context first materializes the base cache and stores it in memory. When another agent later processes the same context, it reuses the stored base cache without recomputation of value projections as illustrated in Figure 2(b). We refer to this scheme as `BaseShared`.

For the adapter output, naively storing it in full-dimension form would largely negate the benefit of sharing, since keeping full-dimension cache per agent in addition to the base

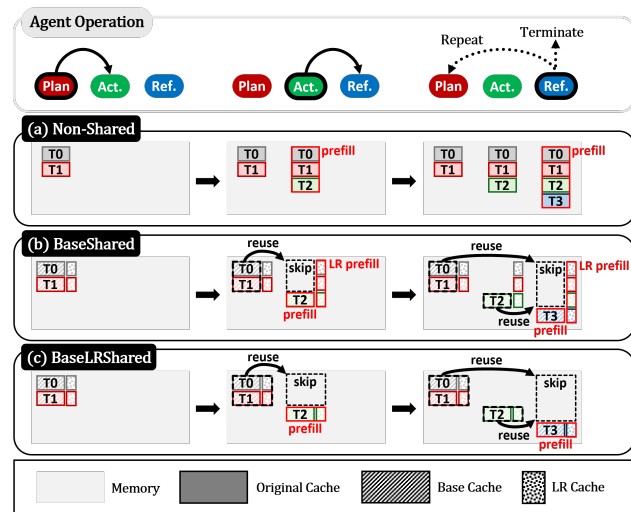

*Figure 2.* Agent iteration and cache accumulation for Non-Shared, `BaseShared`, and `BaseLRShared`. T0 denotes the system prompt shared across agents, and T$i$ denotes trajectory context blocks, formed by concatenating model-generated tokens and retrieved context from external sources. `BaseShared` shares only the base cache and maintains a separate LR cache per agent, whereas `BaseLRShared` shares both the base and LR caches.

cache increases the total cache size to $1 + 1/N$ times that of the default non-shared scheme. Instead, we store the adapter output in its inherent low-rank form as the intermediate output $Y_{\text{lr},i} = X_i A_i$, which we call the LR cache, and reconstruct the required contribution at runtime via the up-projection as $Y_{\text{lr},i} B_i$. In addition, since key cache similarity is sufficiently high across agents, we fully share the key cache. While the same idea can also be applied to the key projection when LoRA is used, LoRA is typically applied to the query and value projections for the best accuracy, so we focus on the value projection in our main implementation. Further accuracy and efficiency analysis for LoRA applied to the key projection is presented in Appendix D.1.

Figure 3 illustrates a forward pass that agent $j$ processes a context segment of length $L_c$ given a prefilled context of length $L_p$ by agent $i$. Here, the LR cache is accumulated over the sequence, and later expanded into the full-dimension adapter contribution via the up-projection $B_i$, then added to the base cache. In terms of mem-

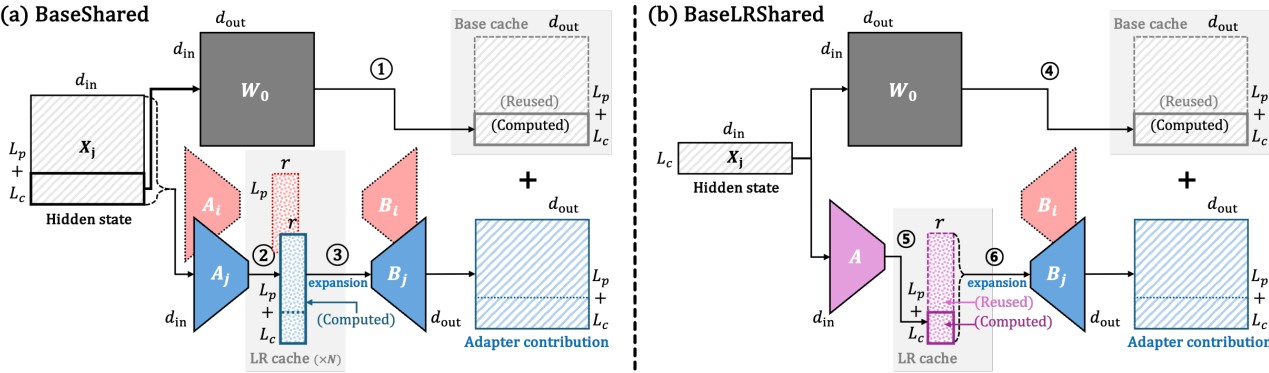

*Figure 3.* Diagram of base and LR cache computation with an initial context of length $L_p$ prefilled by agent $i$, followed by an additional context of length $L_c$ processed by agent $j$, under (a) `BaseShared` and (b) `BaseLRShared`. `BaseShared` maintains per-agent LR caches and computes the LR cache using hidden states for all context tokens not yet processed by the current agent, whereas `BaseLRShared` shares a single LR cache and uses hidden states only for newly appended tokens. Both methods first compute the base cache from the pretrained weights (①, ④). They then compute the LR cache via the LoRA down-projection (②, ⑤), and later expand it to the full dimension via the LoRA up-projection over the full sequence (③, ⑥). Efficient LR cache expansion is described in Section 3.3.

ory, `BaseShared` maintains a single shared base cache along with lightweight per-agent LR caches. Because each LR cache is smaller than the full cache by a factor of $r/d_{\text{out}} \ll 1$, the total KV cache size is reduced to $1/N + r/d_{\text{out}} \simeq 1/N$ of the non-shared scheme. However, in terms of computation, since only the base cache is shared, switching agents requires constructing the LR cache for the entire accumulated context that the new agent has not yet processed; we refer to this as the 'LR prefill' process, as illustrated in Figure 2(b). For example, in Figure 3(a), the hidden states span a sequence of length at least $L_p + L_c$, and the LR cache for agent $j$ must be newly computed via the down-projection $A_j$. At this step, computation of the key and value projection that produces the shared base cache for context segment $L_p$ can be skipped, but the majority of computation (e.g., proceeding MLP layers after the Attention layers) is still required. As a result, the amount of computation remains similar to the non-shared setting, scaling as $O(NL^2 d_{\text{model}})$, where $L$ is the total trajectory length and $d_{\text{model}}$ is the model hidden dimension. We note that this is consistent with prior KV cache sharing methods including DroidSpeak, since selective recomputation requires hidden states for the contexts that were previously processed only by other agents. In summary, `BaseShared` serves as a **robust and memory-efficient solution** applicable to standard multi-LoRA systems. It significantly reduces KV cache memory usage compared to the non-shared baseline while preserving **higher accuracy** than existing prior KV cache sharing methods. As such, `BaseShared` is particularly well-suited for conventional multi-LoRA agents when memory efficiency is a primary concern.

**BaseLRShared** Building on this foundation, we next introduce `BaseLRShared` to achieve **computational acceleration** as well as memory savings. By leveraging the shared-$A$ multi-LoRA architecture, `BaseLRShared` fur-

*Table 2.* Average cosine similarity of the LR cache across agent pairs for the same context in shared-$A$ multi-LoRA architectures.

| Model | (Plan, Action) | (Action, Reflect) | (Reflect, Plan) |
|---|---|---|---|
| LLaMA-3.1-8B-Instruct | 0.9486 | 0.9634 | 0.9607 |
| Ministral-8B-Instruct | 0.9473 | 0.9526 | 0.9498 |

ther eliminates the redundant prefill computation, enabling substantially higher throughput and lower latency than prior approaches. As discussed in Section 2.1, task-specific differences in multi-LoRA systems mainly arise from the up-projection matrices $B_i$, making it effective to share $A$ for improving both parameter efficiency and accuracy (Tian et al., 2024; Yang et al., 2025c). In this setting, we further observe that the LR cache $AX_i$ produced by the shared down-projection can also be shared across agents. As shown in Table 2, the LR cache in shared-$A$ multi-LoRA exhibits high cosine similarity across agents, analogous to the base cache, which motivates sharing the LR cache as well. We therefore maintain a single base cache and a single LR cache for the entire system, and construct agent-specific outputs via their task-specific up-projections $B_i$.

Here, the memory usage is reduced by a factor of $1/N + r/(Nd_{\text{out}}) \simeq 1/N$ relative to the non-shared implementation. Moreover, because both the base cache and the LR cache are available for all previously seen tokens in `BaseLRShared`, an agent does not require recomputation over context processed by previous agents to construct either cache or the hidden states. For example, in Figure 3(b), the base and LR caches for the $L_p$ tokens that are already available, so the forward pass only needs to compute activations for newly appended $L_c$ contexts. Therefore, across $N$ agents over the length-$L$ trajectory, `BaseLRShared` avoids the $N$ separate prefills required by the non-shared implementation. As a result, the overall computational complexity becomes even comparable to full cache sharing, scaling as

**Algorithm 1** Flash-LoRA-Attention Forward (head-wise)

**Require:** $L = L_\mathrm{p} + L_\mathrm{c}, Q \in \mathbb{R}^{L_\mathrm{c} \times d_\mathrm{head}}$
**Require:** $K \in \mathbb{R}^{L \times d_\mathrm{head}}$, base cache $V_\mathrm{base} \in \mathbb{R}^{L \times d_\mathrm{head}}$
**Require:** LR cache $V_\mathrm{lr} \in \mathbb{R}^{L \times r}$, LoRA $B \in \mathbb{R}^{r \times d_\mathrm{head}}$
**Require:** Key, Value block size $B_c$, $T_c = \lceil L/B_c \rceil$
**Require:** Query block size $B_r$, $T_r = \lceil L_\mathrm{c}/B_r \rceil$
**Ensure:** Attn output block $O \in \mathbb{R}^{L_\mathrm{c} \times d_\mathrm{head}}$
1:  $Q, O \to T_r$ blocks $Q_i, O_i$ ($i = 0, \dots T_r - 1$).
2:  $K, V_\mathrm{base} \to T_c$ blocks $K_j, V_{\mathrm{base},j}$ ($j = 0, \dots T_c - 1$).
3:  $V_\mathrm{lr} \to T_c$ blocks $V_{\mathrm{lr},j}$ ($j = 0, \dots T_c - 1$).
4:  **for** $0 \le i < T_r$ **do**
5:      Initialize $O_i \leftarrow 0 \in \mathbb{R}^{B_r \times d_\mathrm{head}}$, $O_{\mathrm{lr},i} \leftarrow 0 \in \mathbb{R}^{B_r \times r}$
6:      Initialize $m_i \leftarrow -\infty \in \mathbb{R}^{B_r}$, $\ell_i \leftarrow 0 \in \mathbb{R}^{B_r}$
7:      Load $Q_i$ to ShrMem
8:      **for** $0 \le j < T_c$ **do**
9:          Load $K_j, V_{\mathrm{base},j}, V_{\mathrm{lr},j}$ to ShrMem
10:         $S \leftarrow \mathrm{Mask}(Q_i K_j^\top / \sqrt{d_\mathrm{head}})$
11:         $m_i^\mathrm{new} \leftarrow \max(m_i, \mathrm{rowmax}(S))$
12:         $\alpha \leftarrow \exp(m_i - m_i^\mathrm{new})$
13:         $P_i \leftarrow \exp(S - m_i^\mathrm{new})$
14:         $\ell_i \leftarrow \alpha \odot \ell_i + \mathrm{rowsum}(P_i)$
15:         $O_i \leftarrow \alpha \odot O_i + P_i V_{\mathrm{base},j}$
16:         $O_{\mathrm{lr},i} \leftarrow \alpha \odot O_{\mathrm{lr},i} + P_i V_{\mathrm{lr},j}$
17:         $m_i \leftarrow m_i^\mathrm{new}$
18:     **end for**
19:     $O_i \leftarrow O_i + O_{\mathrm{lr},i} B$
20:     Write $O_i \leftarrow O_i / \ell_i$
21: **end for**

$O(L^2 d_\mathrm{model})$, considering that the LR cache expansion cost is small (discussed in the Section 3.3).

### 3.3. Flash-LoRA-Attention

Naive runtime expansion of the LR cache introduces non-trivial computational overhead compared to fully shared caching because it scales with both the accumulated sequence length $L$ and the full output dimension $d_\mathrm{out}$. Unlike prior low-rank cache compression methods (Yuan et al., 2023; Tomar et al., 2025; Chang et al., 2025) that treat the expansion of the compressed caches as an inevitable overhead, our approach explicitly reduces it via reordered computation with the attention weight. Specifically, we reorder the matrix multiplications so that the attention-weighted multiplication is performed directly on the LR cache first, and the up-projection is applied afterward, making the overhead scale with the small rank dimension rather than $d_\mathrm{out}$. Concretely, suppose the base cache $V_\mathrm{base} \in \mathbb{R}^{L \times d_\mathrm{out}}$ and a LR cache $V_\mathrm{lr} \in \mathbb{R}^{L \times r}$ for the value projection. Given the up-projection matrix $B$, the adapter contribution is $V_\mathrm{lr} B \in \mathbb{R}^{L \times d_\mathrm{out}}$. A straightforward implementation is to first reconstruct the adapter contribution and then applies attention with attention weights $P$ to produce the attention

output $O$:
$$O = P(V_\mathrm{base} + V_\mathrm{lr} B).$$

This expands the LR cache to the full-dimension $d_\mathrm{out}$ for all $L$ tokens, so the computation overhead scales with both $L$ and $d_\mathrm{out}$. Instead, we exploit associativity and compute

$$O = PV_\mathrm{base} + (PV_\mathrm{lr})B,$$

so the length-$L$ multiplication is performed in the low-rank space, and the multiplication by $B$ is applied afterward. For instance, in a decoding step where the accumulated context length was $L$, the naive implementation adds $\Theta(L\,r\,d_\mathrm{out})$ computation to form $V_\mathrm{lr} B$ already before the attention computation. With reordering, we compute the low-rank intermediate $PV_\mathrm{lr} \in \mathbb{R}^{1 \times r}$ in $\Theta(L\,r)$ and apply the up-projection in $\Theta(r\,d_\mathrm{out})$, for a total of $\Theta(L\,r + r\,d_\mathrm{out})$ computation. As a result, since $L$ is the dominant term that grows over agent trajectories, our approach reduces the computation of LR cache expansion by approximately a factor of $r/d_\mathrm{out} \ll 1$. Algorithm 1 realizes this idea by augmenting FlashAttention (Dao et al., 2022; 2023). Here, $d_\mathrm{head}$ is used instead of $d_\mathrm{out}$ to show the head-wise computation explicit. A generalized analysis of the computation overhead, along with further algorithmic details, is provided in Appendix B.

## 4. Experiments

### 4.1. Implementation Detail

**Agent Setup** We evaluate the accuracy and efficiency of our cache sharing schemes in a multi-hop agent execution framework, following AutoAct (Qiao et al., 2024). We fine-tune three role-specific agents, *plan*, *action*, and *reflect*. The plan agent performs reasoning and selects which external tool to invoke based on the reasoning. The action agent produces tool-specific arguments and executes the selected API calls, including web search (Google Developers, 2025) and Wikipedia lookup (Yao et al., 2023b). The reflect agent reviews the accumulated trajectory and decides whether to terminate with a final answer or to continue the interaction.

**Models and Datasets** We evaluate on LLaMA-3.1-8B-Instruct and Ministral-8B-Instruct. We fine-tune and evaluate on a split of 2.5k HotpotQA (Yang et al., 2018) and 2.0k ScienceQA (Lu et al., 2022) datasets, which require external knowledge and multi-step reasoning to answer. For training, we use the synthetic and filtered agent trajectories released by AutoAct. For evaluation, we run multi-hop inference on the test set with three difficulty levels of HotpotQA and ScienceQA questions, using 20 iterations for each level. Agent prompts and trajectory templates are provided in the Appendix C.1.

**Training Settings** For shared-$A$ multi-LoRA, we simplify the dynamic token-wise routers used in prior work (Tian

*Table 3.* Benchmark accuracy (%) of the default non-sharing scheme and cache sharing schemes on HotpotQA and ScienceQA at each level. For each model, the tiny value in the Avg. column denotes the difference from the corresponding `Non-Shared` baseline.

| Model | Method | HotpotQA | | | | ScienceQA | | | |
|---|---|---|---|---|---|---|---|---|---|
| | | Easy | Medium | Hard | Avg. | 1-4 | 5-8 | 9-12 | Avg. |
| LLaMA-3.1-8B-Instruct | Non-Shared | 42.80 | 41.95 | 31.90 | 38.88 0.00 | 70.63 | 60.54 | 76.75 | 69.31 0.00 |
| | FullShared | 41.15 | 39.15 | 28.90 | 36.40 -2.48 | 68.00 | 55.67 | 72.00 | 65.22 -4.08 |
| | DroidSpeak | 40.60 | 39.55 | 30.15 | 36.77 -2.12 | 68.79 | 59.25 | 74.42 | 67.49 -1.82 |
| | BaseShared | 42.70 | 41.95 | 31.15 | 38.60 -0.28 | 70.38 | 60.75 | 76.58 | 69.24 -0.07 |
| | BaseLRShared | 42.40 | 40.80 | 30.55 | 37.92 -0.97 | 70.42 | 60.25 | 76.71 | 69.13 -0.18 |
| Ministral-8B-Instruct | Non-Shared | 41.30 | 37.75 | 28.75 | 35.93 0.00 | 71.50 | 63.83 | 70.92 | 68.75 0.00 |
| | FullShared | 39.60 | 33.95 | 24.80 | 32.78 -3.15 | 68.83 | 57.33 | 64.25 | 63.47 -5.28 |
| | DroidSpeak | 40.85 | 35.65 | 26.95 | 34.48 -1.45 | 68.88 | 59.54 | 69.96 | 66.13 -2.63 |
| | BaseShared | 40.95 | 37.60 | 29.00 | 35.85 -0.08 | 70.75 | 63.25 | 70.25 | 68.08 -0.67 |
| | BaseLRShared | 41.10 | 37.70 | 27.15 | 35.32 -0.62 | 69.71 | 62.08 | 70.17 | 67.32 -1.43 |

et al., 2024) into a static assignment that selects agent-specific LoRA weights for each predefined role. Under this setting, we observe an accuracy gain over naive multi-LoRA, both with and without cache sharing methods even in mixed domain settings (see Appendix C.2 and C.3). We therefore use the shared-$A$ weights for our main accuracy evaluations. Since shared-$A$ is implemented by duplicating the same $A$ weights across agents, it does not change the multi-LoRA architecture, and therefore does not affect efficiency comparisons. We applied LoRA with rank $r = 8$ on query and value projections. Accuracy and efficiency results for LoRA applied to the query, key, value, and output projections are provided in Appendix D.1, where our setting exhibits dominant accuracy. Detailed training hyperparameters and loss curves are reported in the Appendix C.4.

**Baselines** We compare against the default `Non-Shared` baseline, and several cache sharing methods. These include `FullShared`, which shares the entire KV cache across agents without any recomputation, and DroidSpeak. For DroidSpeak, we follow the official Pareto-optimal configuration in in (Liu et al., 2026) and recompute the top 33% most sensitive layers at runtime, selected by probing HotpotQA accuracy. The selected layers are listed in the Appendix C.5. We note that prefix-aware positional embedding matching methods such as KVLink, KVFlow, and KVComm reduce to `FullShared` in our setup with identical agent prefixes.

**Efficiency Evaluation** We observe that latency in agent systems depends on both the cache sharing method's efficiency and the system accuracy, since lower-accuracy methods tend to take more steps and accumulate longer contexts. To measure the intrinsic efficiency of each cache sharing scheme, we evaluate latency under a controlled trace of sequence lengths, similar to the evaluation protocol of KVComm (Ye et al., 2025). We construct this trace by varying the amount of context retrieved from external tools from 1k to 64k tokens, and we feed the same trace to all schemes. This yields total sequence lengths ranging from 2k to 66k tokens. The

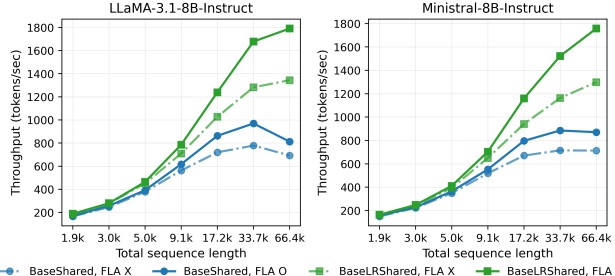

*Figure 4.* System throughput (tokens per second) of `BaseShared` and `BaseLRShared`, with Flash-LoRA-Attention (FLA).

detailed construction of the emulated trace and the latency analysis on the actual HotpotQA benchmark are presented in Appendix C.6 and Appendix D.2, respectively. We conducted experiments on a single NVIDIA A6000 48GB GPU.

## 4.2. Benchmark Accuracy

We demonstrate that both of our cache sharing schemes preserve accuracy more effectively than prior baselines, as shown in Table 3. `BaseShared` stays close to `Non-Shared`, with an average accuracy drop of at most 0.7%. `BaseLRShared` also maintains strong accuracy, with an average drop of at most 1.5%. In contrast, `FullShared` and DroidSpeak exhibit larger average drops, up to 5.3% and 2.6%, respectively. Overall, these results indicate that decoupling the cache into a shared base component and a low-rank component is a key factor for robust KV cache sharing, compared with selective recomputation such as DroidSpeak. Additionally, we provide further analysis of out-of-function incident ratios where the system fails to produce an answer, rank ablations, and deviation of scores in Appendix D.3, D.4, and D.5.

## 4.3. System Efficiency

We report system throughput and time-to-first-token (TTFT) in Table 4 and Table 5, respectively. We define system

*Table 4.* System throughput (tokens per second) under each total sequence length of the traces. `OOM` indicates out-of-memory.

| Model | Method | 1.9k | 3.0k | 5.0k | 9.1k | 17.3k | 33.7k | 66.4k |
|---|---|---|---|---|---|---|---|---|
| | Non-Shared | 176.2 | 262.4 | 401.3 | 592.1 | 669.2 | 683.2 | OOM |
| | FullShared | 193.7 | 293.4 | 468.5 | 808.1 | 1246.1 | 1697.6 | 1826.5 |
| LLaMA-3.1-8B-Instruct | DroidSpeak | 182.4 | 263.5 | 412.3 | 633.5 | 844.2 | 931.0 | 813.1 |
| | BaseShared | 169.0 | 257.0 | 392.3 | 617.3 | 862.8 | 969.6 | 823.2 |
| | BaseLRShared | 188.7 | 279.4 | 463.8 | 785.7 | 1239.4 | 1678.1 | 1790.6 |
| | Non-Shared | 158.9 | 231.0 | 361.5 | 541.2 | 610.7 | 638.4 | OOM |
| | FullShared | 169.9 | 251.4 | 420.3 | 711.2 | 1163.9 | 1538.6 | 1768.3 |
| Ministral-8B-Instruct | DroidSpeak | 160.9 | 251.4 | 360.3 | 570.2 | 785.1 | 856.0 | OOM |
| | BaseShared | 157.0 | 227.4 | 364.0 | 552.1 | 796.5 | 885.0 | 870.5 |
| | BaseLRShared | 164.1 | 247.9 | 410.9 | 703.2 | 1159.2 | 1521.9 | 1757.0 |

*Table 5.* TTFT (second) under each total sequence length of the traces. `OOM` indicates out-of-memory. Lower is better.

| Model | Method | 1.9k | 3.0k | 5.0k | 9.1k | 17.3k | 33.7k | 66.4k |
|---|---|---|---|---|---|---|---|---|
| | Non-Shared | 1.94 | 2.55 | 3.72 | 6.79 | 16.38 | 38.87 | OOM |
| | FullShared | 1.13 | 1.28 | 1.63 | 2.40 | 4.19 | 9.13 | 23.28 |
| LLaMA-3.1-8B-Instruct | DroidSpeak | 1.62 | 2.17 | 3.22 | 5.55 | 11.15 | 25.43 | 67.80 |
| | BaseShared | 1.62 | 2.12 | 3.06 | 5.26 | 10.51 | 23.91 | 67.80 |
| | BaseLRShared | 1.13 | 1.28 | 1.64 | 2.43 | 4.24 | 9.19 | 23.35 |
| | Non-Shared | 2.02 | 2.65 | 3.85 | 6.84 | 17.67 | 41.67 | OOM |
| | FullShared | 1.19 | 1.35 | 1.69 | 2.50 | 4.37 | 9.30 | 20.84 |
| Ministral-8B-Instruct | DroidSpeak | 1.65 | 2.22 | 3.28 | 5.69 | 11.53 | 26.71 | OOM |
| | BaseShared | 1.66 | 2.19 | 3.17 | 5.43 | 10.85 | 25.57 | 59.62 |
| | BaseLRShared | 1.20 | 1.35 | 1.71 | 2.53 | 4.42 | 9.38 | 20.98 |

throughput as the total processed sequence length divided by the end-to-end latency. TTFT is the sum of prefill latencies across agent steps, since each step incurs a new model generation in multi-hop scenarios. We first demonstrate the impact of Flash-LoRA-Attention in Figure 4. It yields up to a 1.24× gain in throughput for `BaseShared` and up to a 1.35× gain for `BaseLRShared`. This shows that reducing LR cache expansion overhead enables substantial speedups. With Flash-LoRA-Attention enabled, `BaseShared` achieves up to a 1.42× gain and `BaseLRShared` achieves up to a 2.46× gain in throughput, approaching the upper bound of full cache sharing with `FullShared`. DroidSpeak reaches a similar throughput gain to `BaseShared`, up to 1.36×, since both methods compute hidden states for tokens that have not been processed by the current agent. A similar trend holds as the context overlap ratio varies, where our methods consistently exhibit high throughput gain (see Appendix D.6). For TTFT, `BaseShared` and `BaseLRShared` provide up to 1.63× and 4.44× reductions, respectively, both exceeding DroidSpeak which achieves up to a 1.56× reduction. `BaseLRShared` achieves TTFT reductions close to those of `FullShared`.

Additionally, `BaseShared` and `BaseLRShared` reduce KV cache memory by nearly 1/3 compared to `Non-Shared` baseline, as shown in Figure 5, which is comparable to other cache-sharing baselines and only marginally higher than `FullShared` within 1GB. This is because the LR caches in `BaseShared` and

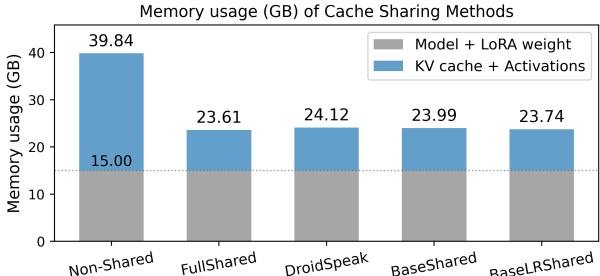

*Figure 5.* Memory usage (GB) of cache sharing methods on total sequence length of 66.4k on Ministral-8B-Instruct.

`BaseLRShared` are negligible in size. We note that under group-query attention (GQA), the hidden state cache used by DroidSpeak is slightly larger than the KV cache for a layer, which led to near out-of-memory (OOM) behavior in some cases of our experiments. We provide a detailed memory analysis in Appendix D.7.

## 5. Conclusion

In this work, we introduce LRAGENT, a KV cache sharing framework for multi-LoRA agent systems that decouples the value cache into a shared base cache and an adapter-dependent LR cache. `BaseShared` reduces KV memory by sharing the base cache, and `BaseLRShared` further reduces computation by sharing the LR cache under shared-*A* multi-LoRA variants while preserving role-specific behaviors. We validate that these methods preserve accuracy close to the non-shared baseline. Flash-LoRA-Attention pro-

vides substantial efficiency gains by avoiding full-dimension materialization of the LR cache, enabling throughput and TTFT improvements close to fully shared caching. Overall, LRAGENT consistently outperforms prior cache sharing baselines in both accuracy and efficiency.

## Acknowledgments

This work was supported in part by Institute of Information & communications Technology Planning & Evaluation (IITP) grant funded by the Korea government (MSIT) (No.RS-2026-25507427: Development of Efficient Architectures and Training Techniques for High-Performance Lightweight AI Models, No.RS-2025-02273157: Development of Low Power Training/Inference Technologies based on AI Semiconductors, RS-2023-00256081: artificial intelligence semiconductor support program to nurture the best talents), and BK21 FOUR program. (Corresponding Author: Jae-Joon Kim).

## Software and Data

We provide a file upload that reproduces the main results in this paper, including training, evaluation, and latency measurements under the same experimental settings. Detailed descriptions and step-by-step guidelines, such as environment setup and commands to run each experiment, are included in the uploaded file.

## Impact Statement

This paper presents work whose goal is to advance the field of deep learning and large language models. There are many potential societal consequences of our work, none of which we feel must be specifically highlighted here.

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

# A. Base Cache and Adapter Output

## A.1. Cache L1 norm

As discussed in Section 3.1, the output contributions from the pretrained base weights and the low-rank adapters differs in magnitude. We analyze the average output magnitude of the value projection in a multi-LoRA system with three agents, where the pretrained base-weight contribution is treated as the base cache and the LoRA contribution is treated as the adapter output. As shown in Figure 6, the base cache and adapter output magnitudes follow similar trends across layers, but the adapter outputs are much smaller, by factors of 27.3 and 14.77 for LLaMA-3.1-8B-Instruct and Ministral-8B-Instruct, respectively.

Given that the base cache remains highly similar across agents while the adapter outputs are largely decorrelated, the adapter output can be viewed as a small but non-trivial, approximately random perturbation to the base cache. This motivates sharing the base cache rather than the full cache.

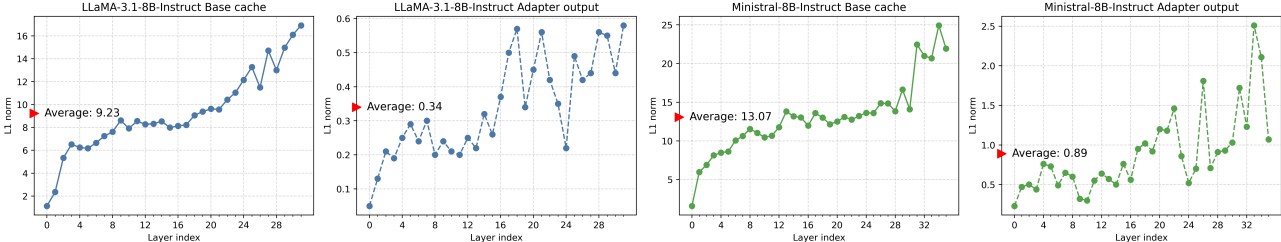

*Figure 6.* L1 norm of the base cache and adapter output across model layers.

## A.2. Input Activation Cosine Similarity

We observe consistently high similarity in input activations across agents, which naturally leads to high base cache similarity.

Figure 7 presents the layerwise cosine similarity of input activations across three agents. The activations show high similarity across agents and exhibit a trend consistent with other forms of cache similarity. Specifically, similarity is higher in earlier layers and gradually decreases in deeper layers, aligning with the observations in Section 3.1 and Figure 1.

Since the base cache is computed by projecting activations with shared pretrained weights, its similarity scales with activation similarity. In contrast, the inclusion of adapter weights in the full KV cache introduces additional variations, leading to lower similarity despite identical input activations.

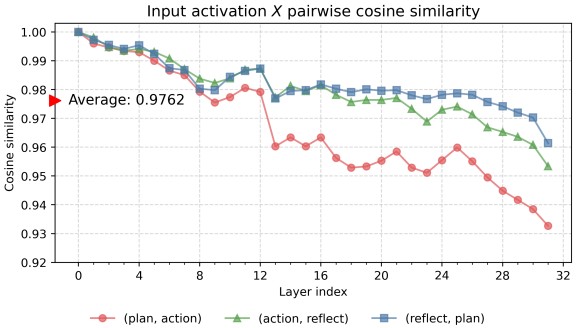

*Figure 7.* Input activation cosine similarity across agent pairs in LLaMA-3.1-8B-Instruct.

## A.3. Cosine Similarity Bound

As shown in Section 3.1, the base cache remains highly similar across agents for the same context, while the adapter outputs are largely decorrelated across agent pairs. This implies that the full cache can be viewed as the base cache perturbed by an approximately random adapter component. In this section, we formalize the resulting effect on similarity: we show that, under the approximate orthogonality assumptions on adapter outputs consistent with our empirical observations, the cosine similarity of the base cache is higher than that of the full cache.

Based on the observed low similarity, or high orthogonality, between the base cache and adapter contributions, as well as among adapter contributions (Tables 1 and 6, respectively), we derive the following bound showing that the full cache exhibits lower cosine similarity. We note that the orthogonality between the base cache and adapter contributions is a practical approximation rather than a strict condition. In multi-LoRA systems with heterogeneous agent roles, the shared pretrained backbone tends to preserve common representations, whereas LoRA fine-tuning captures more role-specific variations. This interpretation is consistent with prior work suggesting that LoRA updates are often only weakly aligned with pretrained directions (Stoica et al., 2025; Luong & Chen, 2026).

Following the multi-LoRA convention in Section 3.1, the base cache and adapter output are defined as:

$$Y_{\text{base},i} := X_i W_0, \qquad \Delta Y_i := X_i \Delta W_i, \qquad Y_i := Y_{\text{base},i} + \Delta Y_i. \tag{4}$$

Assuming that the adapter outputs are approximately orthogonal to the base cache and decorrelated across agents:

$$Y_{\text{base},i}^\top \Delta Y_i = 0, \quad Y_{\text{base},j}^\top \Delta Y_j = 0, \quad Y_{\text{base},i}^\top \Delta Y_j = \Delta Y_i^\top Y_{\text{base},j} = \Delta Y_i^\top \Delta Y_j = 0, \qquad (i \neq j). \tag{5}$$

This satisfies the following:

$$Y_i^\top Y_j = (Y_{\text{base},i} + \Delta Y_i)^\top (Y_{\text{base},j} + \Delta Y_j) = Y_{\text{base},i}^\top Y_{\text{base},j}. \tag{6}$$

Furthermore, we have

$$\|Y_i\|^2 = \|Y_{\text{base},i} + \Delta Y_i\|^2 = \|Y_{\text{base},i}\|^2 + \|\Delta Y_i\|^2 \geq \|Y_{\text{base},i}\|^2, \tag{7}$$

and similarly $\|Y_j\| \geq \|Y_{\text{base},j}\|$. Therefore,

$$\cos(Y_i, Y_j) = \frac{Y_i^\top Y_j}{\|Y_i\| \, \|Y_j\|} = \frac{Y_{\text{base},i}^\top Y_{\text{base},j}}{\|Y_i\| \, \|Y_j\|} \leq \frac{Y_{\text{base},i}^\top Y_{\text{base},j}}{\|Y_{\text{base},i}\| \, \|Y_{\text{base},j}\|} = \cos(Y_{\text{base},i}, Y_{\text{base},j}), \tag{8}$$

and taking expectation over contexts yields

$$\mathbb{E}\big[\cos(Y_{\text{base},i}, Y_{\text{base},j})\big] \geq \mathbb{E}\big[\cos(Y_i, Y_j)\big], \tag{9}$$

which states that the base cache cosine similarity is higher than the full cache cosine similarity.

*Table 6.* Cosine similarity between base cache and adapter outputs across agent pairs.

| Contribution | $\Delta Y_{\text{plan}}$ | $\Delta Y_{\text{action}}$ | $\Delta Y_{\text{reflect}}$ |
|---|---|---|---|
| $Y_{\text{base,plan}}$ | 0.0033 | 0.0050 | -0.0003 |
| $Y_{\text{base,action}}$ | 0.0188 | 0.0985 | 0.0006 |
| $Y_{\text{base,reflect}}$ | -0.0076 | 0.0037 | 0.0488 |

## A.4. Key Cache Cosine Similarity

As noted in Section 3.1, in typical multi-LoRA settings the key cache remains highly similar across agents. In particular, the minimum key cache similarity already exceeds the average base-cache similarity reported in Table 1.

Figure 8 reports pairwise cosine similarity of the key cache for each model. The average similarity is 0.9922 for LLaMA-3.1-8B-Instruct and 0.9840 for Ministral-8B-Instruct, and even the minimum similarity across agent pairs is higher than the corresponding average base-cache similarity: 0.9726 and 0.9530, respectively. This indicates that the primary cross-agent differences come from the value cache, mainly through the adapter-induced component. Therefore, we simply share the entire key cache across agents in all of our schemes.

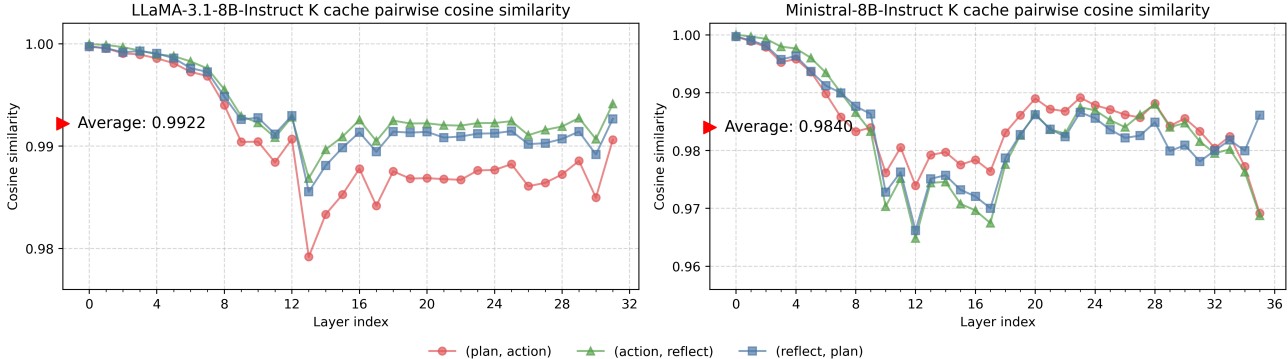

*Figure 8.* K cache Cosine similarity of each agent pairs across the model layers.

## A.5. Agent Role Heterogeneity and Cache Similarity

The observations and experiments in our paper consider strongly heterogeneous agent roles (e.g., planning, action, and reflection), which are functionally distinct in representative multi-agent frameworks (Yao et al., 2023b; Shinn et al., 2023; Qiao et al., 2024). To further examine generalizability under a different role structure, we additionally evaluate debate-style agents on the MATH dataset, following a multi-agent fine-tuning setting (Subramaniam et al., 2025). Table 7 shows that the base cache remains more similar than the full cache, and the LR cache also shows high similarity. These results support our key observation that base cache similarity is consistently higher than full KV cache similarity across different settings.

We also note that our method does not rely on the absolute level of base cache similarity, but on the relative relationship that the base cache is more similar than the full KV cache across agents. Since cross-agent differences mainly come from adapter outputs, increasing agent heterogeneity is expected to reduce full KV cache similarity more than base cache similarity. Our method, LRAgent, is designed to use this property. In contrast, conventional cache sharing methods do not separate the shared pretrained component from the agent-specific adapter component, and thus cannot capture this structure as clearly. Thus in general, we expect our method to exhibit dominant effectivity in various heterogeneous agent role scenarios.

We acknowledge that evaluating on a broader range of datasets would further strengthen the generalizability of our method. However, due to the lack of open-sourced agent trajectories and evaluation frameworks, our experiments are currently limited to AutoAct-based settings. Based on the observations presented above, we nevertheless expect our method to remain effective in diverse heterogeneous scenarios, and we plan to extend our evaluation as more trajectories and benchmarks become available.

*Table 7.* Average cosine similarity between two debate agents (e.g. generation and critic) on the MATH dataset in LLaMA-3.1-8B-Instruct. Results are computed over 128 sampled contexts sequence length of 2k for each.

| Contribution | Full cache | Base cache | LR cache |
|---|---|---|---|
| Cosine Similarity | 0.9553 | 0.9766 | 0.9620 |

## B. Flash-LoRA-Attention

In this section, we provide a detailed description of Flash-LoRA-Attention, which is introduced in Section 3.3, and analyze its computational overhead in a general form, following the notation in Section 3.2 and Figure 3.

**Setup.** We follow the notation in Algorithm 1 and describe the forward pass for a single attention head. Let $L = L_\mathrm{p} + L_\mathrm{c}$, where $L_\mathrm{p}$ is the accumulated context length and $L_\mathrm{c}$ is the context length of the current prefill/decoding step. We denote the query and key as $Q \in \mathbb{R}^{L_\mathrm{c} \times d_\mathrm{head}}$ and $K \in \mathbb{R}^{L \times d_\mathrm{head}}$, and the base value cache as $V_\mathrm{base} \in \mathbb{R}^{L \times d_\mathrm{head}}$. For the LoRA update on the value projection, we save the LR cache $V_\mathrm{lr} \in \mathbb{R}^{L \times r}$ and keep the up-projection matrix $B \in \mathbb{R}^{r \times d_\mathrm{head}}$, where $r \ll d_\mathrm{head}$. With attention weight $P = \mathrm{softmax}(QK^\top/\sqrt{d_\mathrm{head}}) \in \mathbb{R}^{L_\mathrm{c} \times L}$, the output is

$$O = P(V_\mathrm{base} + V_\mathrm{lr}B) = O_\mathrm{base} + O_\mathrm{lr}, \qquad O_\mathrm{base} = PV_\mathrm{base}, \qquad O_\mathrm{lr} = PV_\mathrm{lr}B \tag{10}$$

**Matrix Multiplication Reordering Based on Associativity.** A straightforward implementation materializes the full-dimension adapter contribution $V_\mathrm{lr}B \in \mathbb{R}^{L \times d_\mathrm{head}}$ for all $L$ tokens and then applies attention:

$$O_\mathrm{lr} = P(V_\mathrm{lr}B). \tag{11}$$

This expands the LR cache to the head dimension over the entire trajectory, so the computation grows with both $L$ and $d_\mathrm{head}$. Instead, we exploit associativity and reorder the computation as

$$O_\mathrm{lr} = (PV_\mathrm{lr})B, \tag{12}$$

so the length-$L$ accumulation is performed in rank $r$, and the head-dimension multiplication by $B$ is applied only once per query block.

**Compute Overhead.** We report multiply-add counts up to constant factors and omit the shared base-attention cost for $O_\mathrm{base}$. Without reordering, we first form $V_\mathrm{lr}B$ and then multiply by $P$:

$$\textbf{W/o reorder:} \quad \underbrace{L\,r\,d_\mathrm{head}}_{V_\mathrm{lr}B} + \underbrace{L_\mathrm{c}\,L\,d_\mathrm{head}}_{P(V_\mathrm{lr}B)} = O(L\,r\,d_\mathrm{head} + L_\mathrm{c}L\,d_\mathrm{head}). \tag{13}$$

With reordering, we first accumulate $M = PV_\mathrm{lr} \in \mathbb{R}^{L_\mathrm{c} \times r}$ and then apply $B$:

$$\textbf{Reorder:} \quad \underbrace{L_\mathrm{c}\,L\,r}_{PV_\mathrm{lr}} + \underbrace{L_\mathrm{c}\,r\,d_\mathrm{head}}_{(PV_\mathrm{lr})B} = O(L_\mathrm{c}L\,r + L_\mathrm{c}r\,d_\mathrm{head}). \tag{14}$$

Since $r \ll d_\mathrm{head}$, reordering replaces the dominant $L_\mathrm{c}L\,d_\mathrm{head}$-scaled expansion with an $L_\mathrm{c}L\,r$-scaled low-rank accumulation, and the $d_\mathrm{head}$-dependent multiplication appears only once after the accumulation.

**Flash-LoRA-Attention Kernel Implementation.** Algorithm 1 implements Eq. (12) by extending FlashAttention with one additional low-rank accumulator. For each query block $Q_i \in \mathbb{R}^{B_r \times d_\mathrm{head}}$, the kernel streams over key/value blocks $(K_j, V_{\mathrm{base},j})$ and computes $S = \mathrm{Mask}(Q_iK_j^\top/\sqrt{d_\mathrm{head}})$, maintaining the online-softmax statistics $(m_i, \ell_i)$. Using the same block-wise weights, it accumulates both $O_i \leftarrow O_i + P_iV_{\mathrm{base},j}$ and the low-rank intermediate $O_{\mathrm{lr},i} \leftarrow O_{\mathrm{lr},i} + P_iV_{\mathrm{lr},j}$, where $O_{\mathrm{lr},i} \in \mathbb{R}^{B_r \times r}$ remains in rank $r$ throughout the streaming pass. After all blocks are processed, the kernel applies a single post multiplication $O_i \leftarrow O_i + O_{\mathrm{lr},i}B$ and then normalizes by $\ell_i$. This preserves FlashAttention's memory-efficient I/O pattern while ensuring that the LR cache expansion computation scales primarily with the rank, directly reducing the runtime overhead in both `BaseShared` and `BaseLRShared`.

# C. Implementation Details

## C.1. Agent Prompts and Trajectory Templates

We present an example agent prompt and trajectory from our multi-LoRA agent system, based on the AutoAct implementation (Qiao et al., 2024). As shown in Figure 9, a trajectory accumulates a predefined system prompt, the user question, and multiple rounds of agent-generated tokens interleaved with rule-based context inserts, such as tool outputs retrieved from external sources. The example is from HotpotQA, and ScienceQA follows the similar template except for an explanation on the additional image caption lookup tool.

The system prompt, which specifies the *thought*, *action*, and *observation* format, is identical for all agents and thus fully shared. As a result, prefix positional alignment based KV cache sharing methods (Yang et al., 2025b; Pan et al., 2025; Ye et al., 2025) reduce to `FullShared` in our setup.

The highlighted parts indicate agent-generated outputs, where agents execute in a predefined order (plan-plan-action). The plan agent first produces reasoning and selects a tool, then the action agent generates the tool arguments. If the selected tool is either Web search API, Wikipedia lookup, or image caption lookup that retrieves a predefined image caption, the retrieved context is appended to the trajectory. If the selected tool is *Finish*, the action agent outputs a final answer and the reflect agent is invoked to decide whether the answer is sufficient or whether another information retrieval round is needed. The reflection step is divided into two stages, and it can override an incorrect *Finish* decision and return control to the plan agent when the retrieved evidence is insufficient. The total number of agent iterations is limited to 45 per question.

We train the agents using filtered trajectories generated by a single LLaMA-2-70B-Chat model, provided by AutoAct.

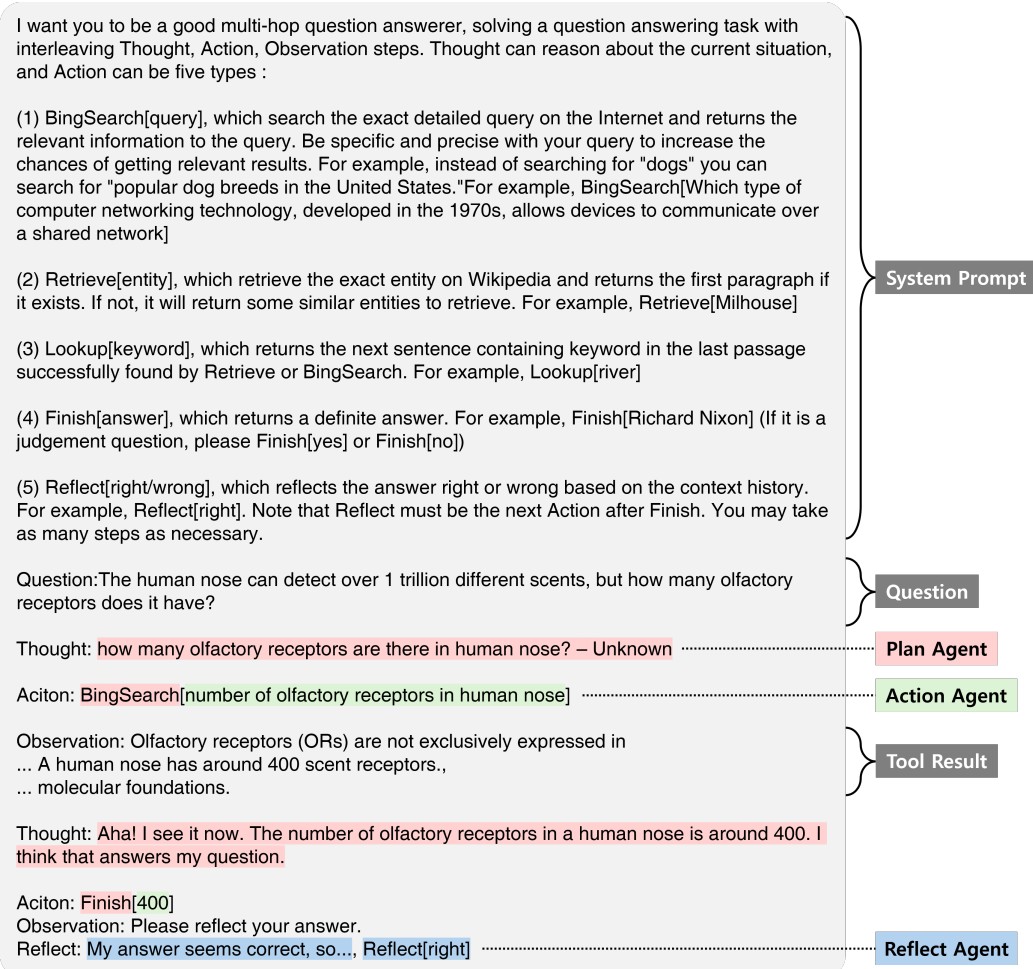

*Figure 9.* Agent prompts and an example of an accumulated trajectory on HotpotQA.

## C.2. Shared-$A$ Multi-LoRA Architecture

We observe that simply sharing the LoRA down-projection matrix $A$ yields higher accuracy than conventional multi-LoRA training with independent $(A_i, B_i)$ pairs, consistent with prior findings (Tian et al., 2024; Yang et al., 2025c). Table 8 reports HotpotQA accuracy with and without shared-$A$. We note that we use the same training conditions and hyperparameters listed in Appendix C.4, which yield the best results in both settings. Across all methods, the shared-$A$ variant improves the original (`Non-Shared`) accuracy and also benefits cache sharing schemes such as `FullShared`, `BaseShared`, and `BaseLRShared`. In particular, `BaseLRShared` degrades when $A$ is not shared, since sharing an LR cache computed with different $A$ matrices and expanding it with a mismatched $B$ introduces large errors.

In our implementation, we duplicate the shared $A$ across agents, which is the same implementation with conventional multi-LoRA architecture and therefore inference efficiency is unchanged. Since shared-$A$ improves accuracy in all settings without introducing change of model structure or inference overheads, we conduct our main experiments using shared-$A$ multi-LoRA trained weights. We note that sharing $A$ also reduces the number of trainable parameters by 33%, providing a efficiency benefit in training.

*Table 8.* Accuracy (%) comparison between non-shared-$A$ and shared-$A$ multi-LoRA variants on HotpotQA easy benchmark.

| Model | Architecture | Non-Shared | FullShared | BaseShared | BaseLRShared |
|---|---|---|---|---|---|
| LLaMA-3.1-8B-Instruct | Non-shared $A$ | 42.05 | 40.30 | 41.85 | 36.25 |
| | Shared-$A$ | 42.80 +0.75 | 41.15 +0.85 | 42.70 +0.85 | 42.40 +6.15 |
| Ministral-8B-Instruct | Non-shared $A$ | 41.10 | 37.40 | 40.80 | 36.95 |
| | Shared-$A$ | 41.30 +0.20 | 39.60 +2.20 | 40.95 +0.15 | 41.10 +4.15 |

## C.3. Shared-$A$ on Multi-Domain Dataset

Previous studies on multi-task LoRA show that sharing the down-projection across domains does not harm model quality and can improve accuracy across tasks (Tian et al., 2024; Yang et al., 2025c).

Following this, we further examine whether the shared-$A$ design in our method remains reliable when training data from different benchmarks are mixed. We conduct an ablation study by mixing trajectories from HotpotQA and ScienceQA and evaluating accuracy on each test set. We use the same hyperparameters and training setup described in Appendix C.4, and train on a shuffled combination of the two datasets. Due to limited availability of open-source agent trajectories, we focus on these two datasets.

As shown in Table 9, training on the mixed dataset does not degrade performance and gives accuracy comparable to training on each dataset separately. The differences are small and within the reported evaluation variance. These results suggest that the shared-$A$ design generalizes well to mixed-task settings without loss in benchmark accuracy.

We acknowledge that, in systems where a multi-LoRA model has already been deployed and the model weights cannot be modified, `BaseLRShared` may not be directly applicable because it requires shared-$A$ to avoid accuracy degradation. However, both our results and prior work suggest that shared-$A$ is an effective design choice when building multi-LoRA systems. From this perspective, we view shared-$A$ not as a limitation, but as an opportunity to improve the multi-task framework while enabling the more efficient `BaseLRShared` scheme.

*Table 9.* Benchmark accuracy (%) of `BaseLRShared` on HotpotQA and ScienceQA subtasks under separate and mixed-dataset training. 'Data Mix' denotes whether trajectories from both datasets are shuffled together during training.

| Data Mix | HotpotQA (Hard) | ScienceQA (9-12) |
|---|---|---|
| x | 31.15 | 76.58 |
| o | 31.30 | 76.65 |

## C.4. Hyperparameter and Loss Curve

We report the training hyperparameters and loss curves for multi-LoRA agents in Table 10 and Figure 10, respectively. Most hyperparameters, including the optimizer, scheduler, and weight decay, follow AutoAct (Qiao et al., 2024), and we perform a grid search over learning rates and the number of training epochs. The sum of training time across all agents is 3.9 hours for HotpotQA and 3.4 hours for ScienceQA on a single 48GB NVIDIA A6000 GPU. We also note that the HotpotQA and ScienceQA test sets consist of 300 and 360 questions, respectively, and we run 20 iterations for each accuracy evaluation.

*Table 10.* Hyperparameter settings for multi-LoRA training.

| Hyperparameter | LLaMA-3.1-8B-Instruct | Ministral-8B-Instruct |
|---|:---:|:---:|
| Optimizer | AdamW | |
| Batch Size | 32 | |
| LR Scheduler | cosine | |
| Max Sequence Length | 32786 | |
| Epochs | 10 | |
| Warmup Ratio | 0.05 | |
| Weight Decay | 0 | |
| Rank | 8 | |
| LoRA Dropout | 0.05 | |
| LoRA Scale | 16 | |
| Learning Rate | Plan: 5e-5
Action: 6e-5
Reflect: 6e-5 | Plan: 5e-5
Action: 9e-5
Reflect: 9e-5 |

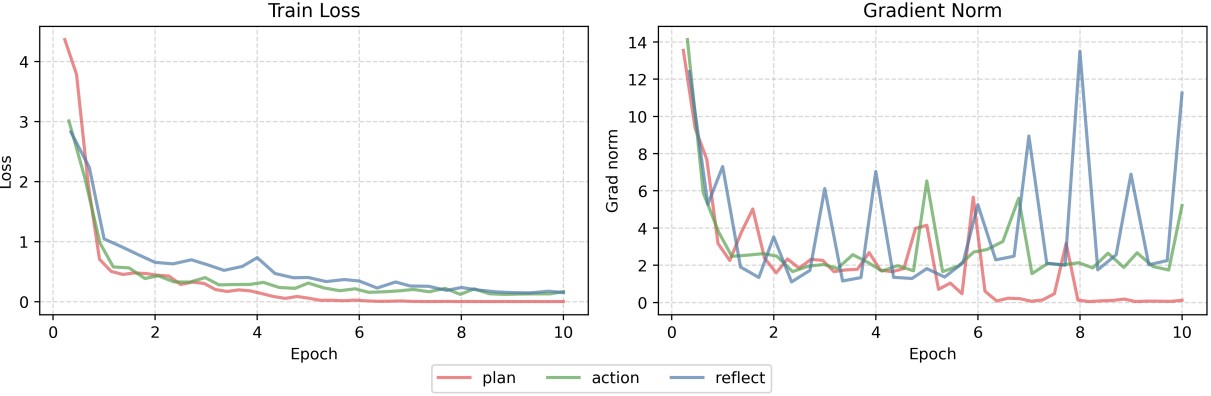

*Figure 10.* Train loss and L2 norm of the gradient update for each agent types.

## C.5. DroidSpeak Recomputation Layer Selection

We implement DroidSpeak as a baseline, following the methodology described in (Liu et al., 2026). In the original implementation, KV caches are selectively recomputed for a set of critical layers, identified by probing the benchmark accuracy drop when the KV cache is directly reused in each layer while the remaining layers are recomputed. DroidSpeak provides a Pareto-optimal configuration that balances accuracy degradation and the inference efficiency gains from cache sharing, corresponding to recomputing 33% of the model layers. Following this guideline, we probe critical layers on HotpotQA and select 11 layers for LLaMA-3.1-8B-Instruct (32 layers total) and 12 layers for Ministral-8B-Instruct (36 layers total). In addition, since the first layer typically does not require recomputation in Ministral-8B-Instruct, we enable hidden state caching that can be passed directly from the previous model to the current model to eliminate computation for these layers. We note that recent models often use group-query attention (GQA), where the hidden state dimension is typically four times larger than the output dimension of the key or value projections, so the hidden state cache can be roughly twice as large as the KV cache of a single layer. This additional memory overhead is discussed in Appendix D.7.

*Table 11.* Selected layers for recomputation in DroidSpeak.

| Model | Selected layers |
|---|---|
| LLaMA-3.1-8B-Instruct | 0, 2, 16, 19, 20, 22, 23, 24, 26, 30, 31 |
| Ministral-8B-Instruct | 1, 4, 5, 12, 14, 15, 17, 21, 22, 25, 29, 31 |

## C.6. Emulated Trace for Efficiency Analysis

In the experiments on HotpotQA or ScienceQA benchmarks, methods with lower accuracy tend to produce longer trajectories and thus accumulate longer contexts. Therefore, to enable fair efficiency comparisons of only the cache sharing method itself under the same context length, we construct a fixed trace of context lengths and an agent schedule. This trace is based on profiled trajectories, including the concatenated context length at each agent step and the number of steps per iteration. On average, each iteration consists of 17 steps, comprising five plan-plan-action cycles and two reflect steps at the end of the iteration.

We vary the retrieved context length $L_{\text{ctx}}$ from 0.25k to 16k, which results in total trajectory lengths ranging from 1.9k to 66.4k, as reported in Table 4 and Table 5. We note that the detailed agent trajectory templates are described in Appendix C.1.

*Table 12.* Agent iteration trace with prefill and generation lengths.

| Agent Type | Step | Prefill | Generation |
|---|---|---|---|
| plan | 1 | 512 | 32 |
| plan | 2 | 8 | 8 |
| action | 3 | 8 | 8 |
| plan | 4 | $L_{\text{ctx}}$ | 32 |
| plan | 5 | 8 | 8 |
| action | 6 | 8 | 8 |
| plan | 7 | $L_{\text{ctx}}$ | 32 |
| plan | 8 | 8 | 8 |
| action | 9 | 8 | 8 |
| plan | 10 | $L_{\text{ctx}}$ | 32 |
| plan | 11 | 8 | 8 |
| action | 12 | 8 | 8 |
| plan | 13 | $L_{\text{ctx}}$ | 32 |
| plan | 14 | 8 | 8 |
| action | 15 | 8 | 8 |
| reflect | 16 | 32 | 32 |
| reflect | 17 | 8 | 8 |

# D. Ablative Experiments

## D.1. Ablation on LoRA Application

Although LoRA is most commonly applied to the query and value projections, which we denote as the `qv` setting, we also evaluate an alternative configuration with the same parameter budget by applying LoRA to the query, key, value, and output projections with rank $r = 4$, which we denote as `qkvo`. Following Section 4.3, we measure HotpotQA accuracy along with system throughput and TTFT under the same emulated trace, where the results are presented in Table 13, 14, and 15.

We find that the non-shared baseline in `qkvo` achieves lower accuracy than in `qv`, and this degradation carries over to all cache sharing methods. Nevertheless, our methods still achieve the best accuracy among the cache sharing approaches, indicating that decoupling the base cache and the LR cache remains effective when LoRA is applied to the key projection.

In terms of system throughput, the `qkvo` setting is inherently less favorable because it introduces additional LoRA computation paths on multiple projections. Moreover, in our schemes, adapter contribution reconstruction from key LR cache must be performed in the head dimension before applying rotary positional embeddings (RoPE), which limits the same associativity-based reordering we exploit for the value cache. Concretely, letting the post-RoPE query be $Q$ and the pre-RoPE key be $K' = K'_{\text{base}} + K'_{\text{lr}}B$ with $K = \text{RoPE}(K')$, the attention score is $QK^\top = Q\big(\text{RoPE}(K')\big)^\top = Q\Big(\text{RoPE}(K'_{\text{base}}) + \text{RoPE}(K'_{\text{lr}}B)\Big)^\top$, so the low-rank reordering from $Q(B^\top K'^\top_{\text{lr}})$ to $(QB^\top)K'^\top_{\text{lr}}$ is not directly applicable because $\text{RoPE}(\cdot)$ applies a position-dependent rotation on the head dimension. As a result, both `BaseShared` and `BaseLRShared` under `qkvo` achieve lower throughput than their `qv` counterparts reported in Table 4. Still, `BaseLRShared` remains more efficient than DroidSpeak. On the other hand, because the adapter output reconstruction primarily affects the generation stage rather than prefill, TTFT under `qkvo` increases marginally overall compared to Table 5. Similarly with the previous results, `BaseLRShared` remains close to `FullShared` in TTFT, while `BaseShared` remains comparable to DroidSpeak.

Overall, our approach maintains the strongest accuracy among cache sharing baselines, and `BaseLRShared` retains a clear efficiency advantage, demonstrating the generality and scalability of our design across LoRA configurations, while `qv` setting used in this paper is favorable across the as mentioned in Section 4.1.

*Table 13.* LLaMA-3.1-8B-Instruct average HotpotQA benchmark accuracy (%) under the `qkvo` scheme.

| Method | Non-Shared | FullShared | DroidSpeak | BaseShared | BaseLRShared |
|---|---|---|---|---|---|
| Accuracy (%) | 38.43 | 34.88 | 36.28 | 38.12 | 37.57 |

*Table 14.* LLaMA-3.1-8B-Instruct system throughput (tokens per second) under each total sequence length of the traces in `qkvo` setting.

| Method | 1.9k | 3.0k | 5.0k | 9.1k | 17.3k | 33.7k | 66.4k |
|---|---|---|---|---|---|---|---|
| Non-Shared | 123.1 | 184.4 | 297.1 | 467.1 | 525.0 | 556.0 | OOM |
| FullShared | 155.8 | 215.1 | 357.2 | 626.3 | 1080.6 | 1544.9 | 1699.4 |
| DroidSpeak | 145.9 | 211.8 | 326.2 | 519.6 | 739.1 | 887.2 | 792.9 |
| BaseShared | 127.2 | 195.6 | 325.7 | 484.9 | 679.3 | 755.6 | 673.2 |
| BaseLRShared | 150.3 | 219.7 | 361.3 | 560.3 | 806.7 | 998.8 | 1051.9 |

*Table 15.* LLaMA-3.1-8B-Instruct TTFT (second) under each total sequence length of the traces in `qkvo` setting.

| Method | 1.9k | 3.0k | 5.0k | 9.1k | 17.3k | 33.7k | 66.4k |
|---|---|---|---|---|---|---|---|
| Non-Shared | 4.79 | 5.00 | 4.94 | 10.52 | 20.73 | 50.04 | OOM |
| FullShared | 1.23 | 1.40 | 1.81 | 2.70 | 4.82 | 9.57 | 24.05 |
| DroidSpeak | 1.77 | 2.29 | 3.37 | 5.84 | 11.54 | 25.43 | 67.80 |
| BaseShared | 1.78 | 2.34 | 3.29 | 5.59 | 10.99 | 26.20 | 70.57 |
| BaseLRShared | 1.27 | 1.46 | 1.92 | 2.89 | 4.93 | 10.30 | 25.30 |

## D.2. Latency on HotpotQA Benchmark

We report the end-to-end system latency on the HotpotQA benchmark. In real-world scenarios, additional latency other than model inference arises from function calls such as web search and Wikipedia retrieval. We therefore split latency into model latency, which includes only the inference (prefill and generation), and end-to-end (E2E) latency, which additionally includes function call latency and the latency of data processing the retrieved context. We also report time-to-first-token (TTFT), defined as the sum of model prefill latencies across agent steps, consistent with Table 5 in Section 4.3.

As shown in Table 16 and Table 17, methods with lower accuracy, such as `FullShared` and DroidSpeak, tend to produce longer sequences and incur higher latency, sometimes even exceeding the `Non-Shared` baseline. This occurs despite their strong efficiency which they have demonstrated in trace-based emulations. These results highlight that overall latency depends not only on the cache sharing efficiency, but also on accuracy. When cache sharing degrades generation quality, the agent is more likely to take additional steps to re-reason and retrieve more external context, which increases sequence length and, in turn, increases E2E latency. Overall, `FullShared` achieves a low TTFT but incurs substantial E2E latency overhead compared to `Non-Shared` on LLaMA-3.1-8B-Instruct. DroidSpeak and `BaseShared` exhibit end-to-end latency similar to `Non-Shared`. `BaseLRShared` achieves the best efficiency while preserving strong accuracy, making it empirically optimal for agentic systems.

*Table 16.* End-to-end (E2E) latency and its breakdown. The lowest latency is highlighted in bold.

| Model | Latency | Level | Non-Shared | FullShared | DroidSpeak | BaseShared | BaseLRShared |
|---|---|---|---|---|---|---|---|
| LLaMA-3.1-8B-Instruct | Model Latency (s) | Hard | 5.90 | 7.39 | 5.75 | 5.97 | **5.70** |
| | | Medium | 5.60 | 6.70 | 5.30 | 5.35 | **5.13** |
| | | Easy | 5.04 | 6.22 | 4.69 | 4.76 | **4.54** |
| | | Avg. | 5.51 | 6.77 | 5.24 | 5.36 | **5.12** |
| | E2E Latency (s) | Hard | 13.63 | 19.73 | 13.75 | 13.73 | **13.59** |
| | | Medium | 12.10 | 18.48 | 11.85 | 12.05 | **11.73** |
| | | Easy | 11.10 | 14.87 | 10.64 | 10.89 | **10.54** |
| | | Avg. | 12.28 | 17.69 | 12.08 | 12.23 | **11.96** |
| | TTFT (s) | Hard | 1.02 | 0.81 | 1.00 | 1.07 | **0.77** |
| | | Medium | 1.01 | 0.77 | 0.98 | 1.02 | **0.66** |
| | | Easy | 1.05 | **0.72** | 0.97 | 1.03 | 0.88 |
| | | Avg. | 1.03 | **0.77** | 0.98 | 1.04 | 0.77 |
| Ministral-8B-Instruct | Model Latency (s) | Hard | 6.64 | 7.23 | 6.82 | 6.79 | **6.39** |
| | | Medium | 6.36 | 6.43 | 6.23 | 6.39 | **6.00** |
| | | Easy | 6.86 | 6.05 | 6.11 | 5.86 | **5.77** |
| | | Avg. | 6.62 | 6.57 | 6.38 | 6.35 | **6.05** |
| | E2E Latency (s) | Hard | 14.65 | 14.36 | 14.98 | 15.06 | **13.91** |
| | | Medium | 12.66 | 14.30 | 12.41 | 12.25 | **11.45** |
| | | Easy | 15.22 | 12.85 | 12.98 | 12.92 | **12.79** |
| | | Avg. | 14.18 | 13.83 | 13.46 | 13.41 | **12.72** |
| | TTFT (s) | Hard | 1.26 | 1.30 | 1.44 | 1.33 | **1.26** |
| | | Medium | 1.17 | **1.07** | 1.26 | 1.17 | 1.12 |
| | | Easy | 1.22 | 1.12 | 1.24 | 1.20 | **1.08** |
| | | Avg. | 1.22 | 1.16 | 1.31 | 1.23 | **1.15** |

*Table 17.* Average of total sequence length (tokens) accumulated during multi-agent execution.

| Model | Level | Non-Shared | FullShared | DroidSpeak | BaseShared | BaseLRShared |
|---|---|---|---|---|---|---|
| LLaMA-3.1-8B-Instruct | Hard | 1099 | 1584 | 1223 | 1039 | 1302 |
| | Medium | 1092 | 1475 | 1108 | 996 | 1229 |
| | Easy | 1088 | 1483 | 1134 | 1077 | 1154 |
| | Avg. | 1093 | 1514 | 1155 | 1038 | 1228 |
| Ministral-8B-Instruct | Hard | 1120 | 1460 | 1355 | 1202 | 1468 |
| | Medium | 1019 | 1398 | 1347 | 1144 | 1406 |
| | Easy | 1154 | 1393 | 1414 | 1142 | 1363 |
| | Avg. | 1098 | 1417 | 1372 | 1162 | 1412 |

## D.3. Out-of-Function Ratio

We define cases where the agent system fails to produce an answer before reaching the maximum number of iterations (e.g., 45) as out-of-function (OOF). In the benchmark accuracy evaluation, these cases are counted as incorrect. However, from a user-experience perspective, returning no answer can be qualitatively different from returning an incorrect answer, and may be considered a more severe failure. We therefore report OOF incidents in addition to benchmark accuracy.

As shown in Table 18, which reports the OOF ratio and its difference from the `Non-Shared` baseline, methods with lower accuracy generally exhibit higher OOF ratios. Consistent with the accuracy results where `BaseShared` and `BaseLRShared` achieve the best accuracy among cache sharing methods, our schemes also yield lower OOF ratios in most cases, except for Ministral-8B-Instruct on ScienceQA.

*Table 18.* Out-of-function (OOF) rate (%) for each benchmark and difficulty level. The underlying value in the Avg. column denotes the difference from the corresponding `Non-Shared` baseline. Lower is better.

| Model | Method | HotpotQA | | | | ScienceQA | | | |
|---|---|---|---|---|---|---|---|---|---|
| | | Easy | Medium | Hard | Avg. | 1-4 | 5-8 | 9-12 | Avg. |
| LLaMA-3.1-8B-Instruct | Non-Shared | 1.50 | 1.80 | 1.65 | 1.65 $_{0.00}$ | 0.00 | 0.13 | 0.21 | 0.11 $_{0.00}$ |
| | FullShared | 2.05 | 2.05 | 3.25 | 2.45 $_{+0.80}$ | 0.29 | 2.46 | 0.54 | 1.10 $_{+0.99}$ |
| | DroidSpeak | 2.10 | 3.10 | 5.15 | 3.45 $_{+1.80}$ | 0.58 | 1.29 | 1.71 | 1.19 $_{+1.08}$ |
| | BaseShared | 1.35 | 1.50 | 2.65 | 1.83 $_{+0.18}$ | 0.21 | 1.83 | 0.13 | 0.72 $_{+0.61}$ |
| | BaseLRShared | 1.25 | 1.80 | 2.25 | 1.77 $_{+0.12}$ | 0.29 | 2.50 | 0.25 | 1.01 $_{+0.90}$ |
| Ministral-8B-Instruct | Non-Shared | 3.90 | 5.15 | 5.80 | 4.95 $_{0.00}$ | 0.38 | 0.17 | 0.83 | 0.46 $_{0.00}$ |
| | FullShared | 7.15 | 7.65 | 10.95 | 8.58 $_{+3.63}$ | 4.17 | 2.96 | 4.92 | 4.01 $_{+3.56}$ |
| | DroidSpeak | 7.05 | 9.50 | 8.20 | 8.25 $_{+3.30}$ | 1.79 | 2.17 | 3.33 | 2.43 $_{+1.97}$ |
| | BaseShared | 6.45 | 6.35 | 9.50 | 7.43 $_{+2.48}$ | 2.75 | 1.92 | 3.25 | 2.64 $_{+2.18}$ |
| | BaseLRShared | 4.45 | 6.55 | 7.60 | 6.20 $_{+1.25}$ | 1.83 | 1.67 | 3.38 | 2.29 $_{+1.83}$ |

## D.4. Rank Ablations

Table 19 reports the accuracy of our method, particularly `BaseShared`, on the QA benchmarks across ranks from 1 to 32. There is a noticeable accuracy gain from rank 1 to 8 because agentic operation, including planning for the given question and tool selection, requires precise adaptation to the specific trajectory dataset. However, since the agent-trajectory training data are relatively small and easy to adapt, we observe only marginal accuracy differences for ranks above 8. Therefore, to minimize both training and inference overhead, we use rank $r = 8$ in all experiments. Experiments on ScienceQA shows a similar trend, indicating the choice of rank 8 is task-independent.

*Table 19.* Rank ablation on benchmark accuracy (%) in BaseShared.

| Benchmark | 1 | 2 | 4 | 8 | 16 | 32 |
|---|---|---|---|---|---|---|
| HotpotQA | 31.35 | 35.98 | 37.88 | 38.60 | 38.43 | 38.50 |
| ScienceQA (9-12) | 71.05 | 73.98 | 75.80 | 76.58 | 76.54 | 76.71 |

### D.5. Accuracy Score Deviation

We report the standard deviation of the average accuracy in Table 3 of Section 4.2 for each baseline and our methods in Table 20. We note that all experiments use a single random seed (42), but accuracy can still vary due to subtle non-deterministic characteristics in external tool usage. For each benchmark level, we run 20 iterations. The accuracy gaps between methods are larger than the observed deviations and therefore we see that the comparisons remain reliable.

*Table 20.* Standard deviation of average benchmark accuracy (%) with 20 iterations.

| Dataset | Non-Shared | FullShared | DroidSpeak | BaseShared | BaseLRShared |
|---------|-----------|-----------|-----------|-----------|--------------|
| HotpotQA | 0.16 | 0.32 | 0.21 | 0.19 | 0.24 |
| ScienceQA | 0.20 | 0.45 | 0.25 | 0.26 | 0.31 |

### D.6. Ablation on the Context Overlap Ratio

We provide analysis on the system throughput when the context overlap ratio varies across agents. Since context overlap is the key premise that enables KV cache sharing, lower overlap reduces the amount of reusable cache and gradually makes all cache sharing methods behave more like the non-shared setting. Table 21 reports throughput under different overlap ratios while using the emulation trajectory length of 33.7k. An overlap ratio of 100% corresponds to the fully shared setting used in our main experiments, whereas 0% represents the case where no cross-agent cache can be reused.

As expected, the throughput advantage of KV cache sharing decreases as the overlap ratio becomes smaller and converges toward the non-shared baseline when the overlap ratio approaches 0%. Nevertheless, across all overlap settings, our methods consistently achieve higher throughput than conventional cache sharing baselines such as `DroidSpeak`. Among them, `BaseLRShared` performs best overall, demonstrating the effectiveness of our approach even when the amount of reusable context is reduced.

*Table 21.* Throughput (tokens/s) under varying context overlap ratios in LLaMA-3.1-8B-Instruct with trajectory length of 33.7k.

| Overlap (%) | Non-Shared | FullShared | DroidSpeak | BaseShared | BaseLRShared |
|-------------|-----------|-----------|-----------|-----------|--------------|
| 100 | 683.2 | 1697.6 | 931.0 | 969.6 | 1678.1 |
| 80 | 683.2 | 1385.1 | 859.0 | 879.9 | 1375.6 |
| 60 | 683.2 | 1121.7 | 789.3 | 805.2 | 1108.2 |
| 40 | 683.2 | 930.3 | 742.4 | 752.7 | 925.0 |
| 20 | 683.2 | 784.4 | 703.8 | 706.3 | 764.9 |
| 0 | 683.2 | 682.2 | 677.9 | 677.1 | 681.6 |

### D.7. Memory Usage

We report the memory usage of each method across traces with diverse trajectory lengths on Ministral-8B-Instruct. We note that the pretrained model weights consume 14.95 GB of memory, and the three LoRA weights add 0.11 GB of memory. Beyond these components, KV cache memory becomes severe in long-context scenarios where retrieved contexts accumulate. Since KV cache sharing methods typically maintain a single shared KV cache for three agents and recompute and overwrite it when needed, their memory usage is similar within 1 GB difference overall.

In particular, `FullShared` has the lowest memory usage because it directly reuses the KV cache without additional components. DroidSpeak additionally maintains a hidden state cache. Since the first layer typically does not require recomputation, its hidden states can be transferred directly from the previous model to the current model, eliminating computation for these layers. However, this cache becomes an overhead in modern group-query attention (GQA) models. The hidden state dimension is often about four times larger than the key or value projection dimension, so the hidden state cache can be roughly twice as large as the KV cache of a single layer. We note that the OOM observed in Section 4.3 mainly arises from memory fragmentation, despite the gap between the GPU's peak capacity and the actual allocated usage. For `BaseShared` and `BaseLRShared`, there exists an additional LR cache, which is three times larger in `BaseShared` than in `BaseLRShared`, but it remains negligible due to its small dimension relative to the base cache. As a result, our schemes achieve memory usage close to `FullShared` as well as other cache sharing methods.

*Table 22.* Memory usage (GB) for each total sequence length trace.

| Total Seq. Len. | Non-Shared | FullShared | DroidSpeak | BaseShared | BaseLRShared |
|---|---|---|---|---|---|
| 1.9k | 15.72 | 15.25 | 15.26 | 15.26 | 15.25 |
| 3.0k | 16.10 | 15.38 | 15.40 | 15.40 | 15.39 |
| 5.0k | 16.87 | 15.65 | 15.68 | 15.67 | 15.65 |
| 9.1k | 18.40 | 16.18 | 16.25 | 16.23 | 16.19 |
| 17.3k | 21.46 | 17.24 | 17.37 | 17.34 | 17.27 |
| 33.7k | 27.59 | 19.36 | 19.62 | 19.56 | 19.43 |
| 66.4k | 39.84 | 23.61 | 24.12 | 23.99 | 23.74 |
| 132.0k | 64.34 | 32.11 | 33.12 | 32.87 | 32.37 |

