# OpenReview forum: "LRAgent: Efficient KV Cache Sharing for Multi-LoRA LLM Agents"
_ICML.cc/2026/Conference — ICML 2026 regular_

### Official Review · Reviewer_k9PK · 2026-03-09

**Soundness:** 3
**Presentation:** 2
**Significance:** 3
**Originality:** 2
**Overall Recommendation:** 4
**Confidence:** 4

**Summary:**

This paper proposes "LRAgent", aiming to optimize Key-Value (KV) cache management under the multi-LoRA architecture in multi-agent systems. The authors propose that although LoRA adapters introduce role-specific behaviors, the activation values generated by the frozen Base Model remain highly similar across different agents. The paper proposes decomposing the KV Cache into a shared "Base Cache" and an agent-specific "LR Cache".

**Compliance With Llm Reviewing Policy:**

Affirmed.

**Final Justification:**

I  decide to raise my score. The authors' rebuttal effectively addressed my major concerns. Specifically:
1.Table A resolves my concern about agent heterogeneity by proving the method generalizes well to different agent role structures (e.g., debate agents).
2.Table B (Rank ablation) clearly demonstrates that the performance saturates at rank 8 across different tasks, confirming that the parameter choice is principled rather than cherry-picked.
3.Table C validates the architectural completeness by confirming the framework's applicability to Key projections without performance degradation.
The LRAgent framework and the Flash-LoRA-Attention kernel are now comprehensively verified. I suggest the authors include these tables in the final manuscript.

**Key Questions For Authors:**

1.Supplement relevant comparative and ablation experiments, expand the types of datasets and task scopes, and conduct further experiments and analysis such as mixing multiple types of tasks.

2.Quantify the analysis of multiple similarity metrics for the activation values $X_l$ at different layers between Agent A and Agent B.

3.The peak memory usage and computational overhead when applying the proposed method to existing LoRA models require further analysis.

**Limitations:**

yes

**Strengths And Weaknesses:**

Strengths  of paper：

1.The CUDA kernel proposed in the paper keeps the computation of attention scores within a low-rank space through reordered matrix multiplications. It effectively reduces the memory bandwidth and computational overhead from $O(L \cdot d_{head})$ to $O(L \cdot r)$.

2.Addressing the memory redundancy issue in multi-agent systems aligns with current research hotspots.

3.The proposed method demonstrates significant throughput improvement (2.46x) and memory reduction.

Weaknesses of paper:

1."Cache Similarity" strongly relies on the assumption that "the Adapter output is approximately orthogonal to the Base Cache", which may be too idealized in practical scenarios.

2.Although the BaseShared method supports standard, independently trained LoRA models, it seems to save memory and cannot reduce computational redundancy.

3.The paper specifies that all agents extract features within the same low-dimensional subspace. The three datasets used in the experiments (Plan, Action, Reflect) are highly homogenized, lacking experimental analysis across diverse scenario tasks.

4.The authors used an extremely lightweight rank ($r=8$) in the vast majority of the experiments. In the ablation experiments in Table 17, even when increased to $r=32$, the accuracy showed no significant change. There is a lack of further analysis and experimental demonstration on whether the rank size is task-independent.

5.The approach of sharing the Key Cache was chosen due to its high similarity; however, the results in Appendix D.1 show that applying LoRA to query, key, value, and output yields poor performance, yet the rank also changes. There is a lack of further rigorous ablation evaluation to verify the causes of performance degradation, and whether the proposed method is inapplicable to the Key Cache.

---

> ### Author Rebuttal · Authors · 2026-03-30
>
> Thank you for your constructive feedback. Due to space limitations, we provide concise descriptions with summarized results, and will be happy to discuss further.
>
> >### **Response to Weakness 1**
>
> The approximate orthogonality between adapter outputs and base cache reflects a general property of LoRA fine-tuning, where updates adapt models to previously unseen domains, yielding subspaces weakly aligned with the pretrained weight space. This is supported by prior works [1–2], demonstrating that LoRA updates exhibit weak spectral alignment with dominant pretrained directions, making the assumption broadly valid in practice.
>
> [1] Model merging with SVD to tie the Knots, George Stoica et al.
>
> [2] Why LoRA Fails to Forget: Regularized Low-Rank Adaptation Against Backdoors in Language Models, Hoang-Chau Luong et al.
>
> >### **Response to Weakness 2**
>
> Memory reduction is itself a primary goal in long-context LLM serving, where OOM prevents system execution entirely; accordingly, most KV cache compression works focus on memory reduction over computational cost during prefill. BaseShared follows this line while outperforming prior methods in accuracy and maintaining competitive throughput. Furthermore, we propose BaseLRShared, which also reduces computational cost. **Importantly**, the shared-A design which it adopts is not a constraint since it improves accuracy in both single-dataset and diverse multi-task settings (e.g., HydraLoRA and MTL-LoRA on BBH, GLUE, and CommonsenseQA with 8+ domains).
>
> >### **Response to Weakness 3**
>
> Planning, action, and reflection are established as fundamentally distinct agent roles in smajor multi-agent works (e.g., ReAct, Reflexion, ReWOO, BOLAA, AutoAct), making our setup *heterogeneous* rather than *homogenized*. Furthermore, the shared-A structure generalizes across diverse domains, as matrix B suffices to capture heterogeneity. Additional experiments with a different role structure (debate) and dataset (MATH) following [3] in Table A consistently show high base and LR cache similarity with lower full-cache similarity, confirming the generality of our method.
>
> Table A. Cosine similarity of 2 debate agents.
>
> | Contributions | Full cache | Base cache | LR cache |
> |---|---|---|---|
> | Cos. Sim. | 0.9553 | 0.9766 | 0.9620 |
>
> [3] https://llm-multiagent-ft.github.io
>
> >### **Response to Weakness 4**
>
> Model accuracy increases with LoRA rank and saturates, with additional experiments (ranks 1–2 in Table B and Table 17) confirming saturation at rank 8. This aligns with the LoRA paper, where ranks below 16 mostly achieves sufficient accuracy, making rank 8 an established choice, not an extreme setting. Experiments on ScienceQA (Table B) shows a similar trend, indicating the choice of rank 8 is task-independent.
>
> Table B. Rank ablation on the benchmark accuracy (%) in BaseShared.
>
> | Benchmark | 1 | 2 | 4 | 8 | 16 | 32 |
> |---|---|---|---|---|---|---|
> | HotpotQA | 31.35 | 35.98 | 37.88 | 38.60 | 38.43 | 38.50 |
> | ScienceQA 9-12 | 71.05 | 73.98 | 75.80 | 76.58 | 76.54 | 76.71 |
>
>
> >### **Response to Weakness 5**
>
> Experiments in Table 11 uses a fixed parameter budget (thereby rank 4 for qkvo) for fairness; the non-shared baseline already underperforms qv under this configuration, explaining the performance degradation. Table C further shows that rank-8 LoRA on K projection yields marginal accuracy increase, confirming applicability to K projection without degradation.
>
> Table C. HotpotQA accuracy (%) on LoRA applied to K projection.
>
> | Key Proj. LoRA | BaseShared | BaseLRShared |
> |---|---|---|
> | x (our paper) | 38.60 | 37.92 |
> | o | 38.77 | 37.95 |
>
> >### **Response to Question 1**
>
> Our paper covers diverse evaluations and ablations as detailed in the Appendix. Nevertheless, given limited open-source agent trajectories, we conducted the mixed-task experiment by training on a shuffled HotpotQA and ScienceQA dataset. Table D shows performance comparable to single-dataset case within score deviation, confirming generalization.
>
> Table D. Benchmark accuracy (%) of BaseLRShared on mixed dataset.
>
> | Mixed dataset | HotpotQA Hard | ScienceQA 9-12 |
> |---|---|---|
> | x (our paper) | 31.15 | 76.58 |
> | o | 31.30 | 76.65 |
>
> >### **Response to Question 2**
>
> Due to space constraints, we report layer-averaged input activation cosine similarity in Table E. The similarity is high across agents and follows the same layer-wise trend as the base cache (e.g. decrease in deeper layers), confirming base cache similarity naturally follows from activation similarity.
>
> Table E. Input activation cosine similarity.
>
> | Pair | Cos. Sim. |
> |---|---|
> | (plan, action) | 0.967 |
> | (action, reflect) | 0.980 |
> | (reflect, plan) | 0.982 |
>
> >### **Response to Question 3**
>
> Section 4.3 applies to both non-shared-A and shared-A designs, as matrices A are simply set to identical values within the standard multi-LoRA architecture, introducing no additional bottlenecks. Latency and throughput thus remain consistent between both configurations.

---

> > ### Author Rebuttal · Reviewer_k9PK · 2026-04-03
> >
> > Thanks to the authors for the supplemented experiments, which resolved the Weaknesses I pointed out. The proposed LRAgent framework provides a practical, mathematically sound, and experimentally verified solution to the KV Cache redundancy problem in multi-LoRA Agent systems. I suggest the authors integrate Table A (debate Agent validation), Table B (Rank ablation experiment), and Table C (Key projection ablation experiment) into the main text of the paper.
> > I will raise my score.

---

> > > ### Author Response · Authors · 2026-04-03
> > >
> > > We thank the reviewer for the positive feedback and for acknowledging that the additional experiments address the previously raised concerns. In the revised version, we will integrate the additional experimental results, particularly Tables A, B, and C, into the main section to further strengthen the clarity and completeness of our results.

---

### Official Review · Reviewer_yxXo · 2026-03-12

**Soundness:** 4
**Presentation:** 2
**Significance:** 3
**Originality:** 3
**Overall Recommendation:** 4
**Confidence:** 4

**Summary:**

This paper introduces LRAgent, a novel KV cache sharing framework explicitly designed for Multi-LoRA Large Language Model (LLM) agent systems. The authors observe that in a Multi-LoRA setup, cache differences across agents processing the same context are dominated by small, adapter-specific outputs, while the activations from the shared pre-trained backbone remain highly similar. Leveraging this insight, LRAgent decouples the kv cache into a shared base component and an agent-dependent low-rank (LR) component.

The paper proposes two caching strategies: **BaseShared** and **BaseLRShared** to significantly reduce memory and compute redundancies. Furthermore, to mitigate the computational bottleneck of expanding the LR cache to full dimensions at runtime, the authors develop **Flash-LoRA-Attention**. This is an elegant kernel-level optimization that reorders matrix multiplications using associativity to perform attention directly in the low-rank space. Extensive experiments on agentic QA benchmarks (HotpotQA and ScienceQA) demonstrate that LRAgent achieves throughput and latency close to fully shared caching while maintaining accuracy near the non-shared baseline.

**Compliance With Llm Reviewing Policy:**

Affirmed.

**Final Justification:**

the response has addressed my questions and i maintain my positive rating.

**Key Questions For Authors:**

I do not have specific questions that require a response during the rebuttal phase. Instead, I strongly recommend that the authors carefully polish the presentation based on the points raised in the "Weaknesses" section. Specifically, breaking down the lengthy sentences in the abstract, clarifying the notation shift from tasks to agents, and properly contextualizing the limitations of prior baselines earlier in the text will significantly enhance the reading experience and the overall impact of the paper.

Please note that I deem the current score sufficient to reflect the paper's overall performance. Therefore, I do not intend to adjust the score during the rebuttal period.

**Limitations:**

No. The authors have not explicitly included a dedicated Limitations section discussing the technical drawbacks or constraints of their framework. Adding this section is recommended, as it would further enhance the overall academic quality of the work.

**Strengths And Weaknesses:**

**Strengths:**
* **Originality & Significance:** The paper addresses a highly relevant but under-explored problem. While Multi-LoRA architectures are widely used for agent role specialization, existing KV cache sharing methods largely overlook this specific mathematical structure. By identifying this gap, the authors propose a highly original framework that holds significant practical value for deploying complex, multi-step LLM agent systems.
* **Soundness (Algorithm & Engineering):** The algorithmic intuition is simple yet mathematically sound and highly effective. Decoupling the cache into a shared base and a low-rank adapter component is a clever exploitation of the LoRA structure. More importantly, the engineering execution is exceptionally solid. The introduction of Flash-LoRA-Attention bridges the gap between theoretical memory savings and actual runtime acceleration, demonstrating a deep understanding of hardware-aware optimization.
* **Soundness (Empirical Evaluation):** The experimental results are comprehensive and convincing. The authors rigorously evaluate not just memory savings, but also end-to-end latency, system throughput, and time-to-first-token (TTFT). The accuracy retention compared to selective recomputation baselines (like DroidSpeak) is clearly demonstrated.

**Weaknesses:**
* **Presentation (Abstract Clarity):** The abstract contains overly long and complex sentences that hinder readability. For instance, the sentence explaining the core mechanism spans over 40 words with multiple clauses. Breaking this down would significantly improve the paper's initial impact.
* **Presentation (Notation Ambiguity):** There is a subtle but confusing shift in mathematical notation. In Section 2.1, the index $i$ is explicitly defined as the "task index". However, in Section 3, $i$ seamlessly transitions to represent the "agent" (e.g., agent $i$). While readers can infer the mapping, a formal statement linking tasks to agent roles should be provided for mathematical rigor.
* **Presentation (Contextualizing Prior Work):** In the Introduction and Section 2.2, the authors list prefix-aware cache sharing methods and selective recomputation as prior works. However, they fail to explicitly articulate *why* these methods fall short in the Multi-LoRA setting until much later. The crucial detail that prefix-aware methods simply reduce to a naive **FullShared** baseline because agents share identical system prompts is buried in Section 4.1. This limitation of prior work should be highlighted earlier to strengthen the motivation.
* **Soundness (Limited Baselines):** The empirical comparison mainly relies on a naive **FullShared** approach and **DroidSpeak**. While this is understandable because Multi-LoRA cache sharing is a novel sub-field with few direct competitors, the evaluation would be more robust if the authors included 1-2 naive adaptations of other existing methods (even if their performance is suboptimal) to further prove the absolute necessity of the decoupled approach.

---

> ### Author Rebuttal · Authors · 2026-03-30
>
> Thank you for your constructive comments. We appreciate the insightful questions and feedback you have provided. We will incorporate the following discussion and changes into the revised manuscript to further strengthen the clarity of our approach, if given the opportunity. Please let us know if there are any additional questions or remaining ambiguities.
>
>
> >### **Response to Weakness 1 on Presentation**
>
>
> We acknowledge that the sentences in abstract, particularly lines 24–38 in our paper are overly long. Therefore, we plan to break them down and revise this section as follows:
>
> Based on this observation, we propose LRAgent, a KV cache sharing framework for multi-LoRA agents. It decomposes the cache into two components, a shared base component derived from pretrained weights and an adapter-dependent component derived from LoRA weights.
> LRAgent reduces memory overhead by sharing the base component across agents and storing the adapter component in its inherent low-rank form. It also reduces computational overhead. Enabled by shared-A multi-LoRA architectures, it shares the low-rank cache and avoids redundant computations for contexts that have already been processed by other agents.
>
> >### **Response to Weakness 2 on Presentation**
>
>
> We thank the reviewer for pointing out this ambiguity. We agree that the transition of notation from task index to agent index may be unclear. To address this, we will add a clarifying statement in Section 3 as:
> We note that the index $i$ denotes both the task and the corresponding agent role, assuming a one-to-one mapping between tasks and agents in our setting.
>
> >### **Response to Weakness 3 on Presentation**
>
> We thank the reviewer for this helpful suggestion. We acknowledge that making the limitation of prior prefix-aware KV cache sharing methods explicit in earlier sections would improve the reader’s understanding of prior work.
> To address this, we will revise the Section 1 and Section 2.2 to clearly state that prefix-aware methods rely on differences in input prefixes, and therefore reduce to a naive FullShared baseline when agents share identical system prompts.
>
> >### **Response to Weakness 4 on Soundness**
>
> As noted in the previous point, most existing KV cache sharing methods are designed for prefix diversity and effectively reduce to the FullShared baseline when applied to our setting. To the best of our knowledge, KV cache sharing specifically tailored for multi-LoRA agent systems has not been explored prior to our work. Therefore, we included general KV cache sharing approaches such as DroidSpeak as baselines. We consider this a strong baseline, as it has demonstrated superior accuracy compared to alternatives such as CacheBlend.
> Nevertheless, we acknowledge that incorporating additional adaptations of existing methods such as CacheBlend could further strengthen the empirical analysis. We will consider presenting such baselines to provide a more comprehensive evaluation as soon as possible.

---

> > ### Author Rebuttal · Reviewer_yxXo · 2026-04-01
> >
> > I will maintain my positive score. Please address those promised in the next version.

---

> > > ### Author Response · Authors · 2026-04-08
> > >
> > > We sincerely thank the reviewer for the response and the detailed interest in our work. We will carefully address the remaining points in the revised version.

---

### Official Review · Reviewer_4A6q · 2026-03-12

**Soundness:** 3
**Presentation:** 3
**Significance:** 2
**Originality:** 2
**Overall Recommendation:** 4
**Confidence:** 3

**Summary:**

This paper proposes LRAGENT, a KV-cache sharing framework for multi-agent LLM systems built with multiple LoRA adapters over a shared backbone. The method is based on the observation that, for a shared context, differences across agents mainly arise from LoRA-specific outputs, while backbone activations remain similar. Using this, the paper decomposes cache representations into a shared base part and an adapter-dependent low-rank part, and introduces BaseShared, BaseLRShared, and Flash-LoRA-Attention to improve memory efficiency and reduce redundant computation. Experiments show that the approach preserves performance close to the non-shared baseline while improving throughput, TTFT, and memory usage over prior cache-sharing baselines in the evaluated multi-LoRA agent setting.

**Compliance With Llm Reviewing Policy:**

Affirmed.

**Final Justification:**

The follow-up response improves clarity and better specifies the intended scope of the method, namely multi-agent systems implemented as multiple LoRA adapters over a shared backbone. However, this also makes clear that the method is tied to this restricted setting, and its behavior under broader forms of agent heterogeneity remains unclear. The rebuttal also does not provide a systematic characterization of performance as heterogeneity increases, relying instead on qualitative explanations and limited observations. I therefore keep my score.

**Key Questions For Authors:**

1.  The stronger BaseLRShared variant relies on the shared-A multi-LoRA assumption, motivated by prior observations that A captures more cross-task common structure while B is more task-specific. Could the authors clarify the intended applicability range of this assumption? In particular, do the authors have additional evidence on when shared-A remains effective and when it may become a representational bottleneck as agent roles become more heterogeneous?
2. The method appears fundamentally tied to multi-agent systems implemented as multiple LoRA adapters over the same pretrained backbone. Could the authors clarify how they view the scope of applicability in practice? For example, how would the proposed sharing strategy extend, if at all, to settings where planner, actor, or reflector agents are built on different base models or only partially overlapping architectures?
3. Could the authors provide additional discussion, or ideally analysis, on how the benefits of BaseShared / BaseLRShared change as inter-agent context overlap decreases or as role heterogeneity increases? This would help clarify whether the gains degrade gracefully outside the current benchmark setting.

**Limitations:**

no. The authors should discuss the method’s main limitations more explicitly, especially its reliance on same-backbone multi-LoRA agents and, for BaseLRShared, the shared-A assumption. The impact statement is also quite minimal; even for an efficiency-focused paper, it would be helpful to briefly discuss the positive effects of reduced inference cost as well as the possible downstream implications of making multi-agent LLM systems easier to scale and deploy.

**Strengths And Weaknesses:**

**Soundness**

Strengths:
1. The technical motivation is reasonably grounded. The paper first shows that, under the same context, base-cache representations across agents remain highly similar while adapter outputs are much less aligned, which supports the idea of sharing the base cache while preserving LoRA-specific components separately. The reported cosine-similarity analysis provides reasonable empirical support for the proposed decomposition.

2. The proposed method is internally coherent: the paper defines a decomposition into a shared base cache and a low-rank adapter-dependent cache, and then derives two sharing schemes from it. BaseShared mainly targets memory savings, while BaseLRShared further reduces redundant prefill computation under a shared-A multi-LoRA setting.

3. Flash-LoRA-Attention is not just a vague systems optimization; the paper gives a concrete reordered computation and complexity argument showing why operating in low-rank space before the up-projection reduces LR-cache expansion overhead.

Weaknesses:
1. The key assumption is mostly empirical rather than strongly justified. The method relies on base-cache similarity across agents and, for BaseLRShared, on LR-cache similarity under shared-A. While the paper provides evidence in its evaluated setup, it is not fully clear how robust these assumptions remain when agent roles become more heterogeneous or are trained on more divergent distributions.
2. The stronger BaseLRShared variant depends on the shared-A design, which is a meaningful restriction. The strongest efficiency results therefore do not apply to arbitrary multi-agent systems, but specifically to multi-LoRA agents sharing the same backbone and, in the stronger variant, also sharing the down-projection matrix.

3. While the evaluation is consistent with the paper’s claims in the studied setting, it does not yet establish how performance degrades as inter-role heterogeneity increases or as the agent architecture departs from the same-backbone multi-LoRA assumption.

**Presentation**

Strengths
1. The paper is generally well organized. The progression from empirical observation, to method decomposition, to BaseShared / BaseLRShared, and finally to Flash-LoRA-Attention is easy to follow. Figures 1–3 and the separation between the two sharing schemes help make the design intuitive.
2. The paper does a good job distinguishing the roles of the three components: BaseShared for memory-oriented sharing, BaseLRShared for stronger sharing under shared-A, and Flash-LoRA-Attention for efficient use of low-rank cache during attention.
3. Experimental tables are clear and make the trade-off easy to read: accuracy degradation is small relative to Non-Shared, while throughput and TTFT improve substantially, especially for BaseLRShared.

Weaknesses
1. Some claims are broader than what is directly validated. The framing sometimes suggests a general multi-agent KV-sharing solution, whereas the method and experiments are tailored to a more specific same-backbone multi-LoRA setting. Tightening this wording would improve precision.
2. The paper would benefit from a more explicit discussion of failure modes and applicability boundaries, especially around when shared-A is expected to help versus when it could become a bottleneck.
3. The presentation would also be stronger with a clearer separation between claims that apply to BaseShared in general same-backbone settings and those that additionally depend on the shared-A assumption in BaseLRShared.

**Significance**

Strengths
1. The problem is practically relevant. In multi-agent LLM systems with long shared trajectories, repeated KV construction and redundant prefills are real inference bottlenecks, so reducing both memory and latency is useful. The paper targets this directly rather than only optimizing a toy setup.
2. The measured gains are meaningful. BaseShared and BaseLRShared preserve accuracy much better than FullShared, while BaseLRShared gets very close to FullShared in throughput and TTFT. The reported efficiency improvements appear substantial rather than cosmetic.

Weaknesses

1. The practical impact depends heavily on deployment assumptions. If a real multi-agent system uses different backbones for different roles, or only weakly overlapping contexts, the method becomes less applicable. Thus, the practical significance appears strong for an important class of same-backbone multi-LoRA deployments, but less clearly broad across heterogeneous multi-agent LLM systems.
2. The empirical scope is still somewhat limited: two 8B models and two agentic QA benchmarks are enough to demonstrate promise, but not enough to completely settle how broadly the method transfers to other agent environments or larger-scale production traces.

**Originality**

The paper presents a well-motivated and reasonably original cache-sharing formulation tailored to the multi-LoRA setting, rather than a generic KV-sharing mechanism. Its main originality lies in exploiting the LoRA decomposition to separate a shared backbone cache from low-rank adapter-dependent cache, and in integrating this formulation with an efficient attention implementation. While this is not a new learning algorithm, it is a meaningful methodological contribution to efficient inference for LLM agent systems.

A key caveat is that the stronger BaseLRShared variant relies on the shared-A multi-LoRA assumption, motivated by prior work suggesting that A captures more cross-task common structure while B is more task-specific. Although the paper cites prior work and presents supporting cosine-similarity evidence, it does not clearly characterize when this assumption continues to hold and when it may become a representational bottleneck under strongly heterogeneous agent roles.

---

> ### Author Rebuttal · Authors · 2026-03-30
>
> Thank you for your constructive feedback. Due to space limits, we provide concise descriptions with summarized results, and we are happy to discuss further.
>
> >### **Response to Soundness W1, Soundness W3, Significance W2, and Question 3**
>
> Our method relies not on the *absolute* similarity of the base cache, but on the *relative* relationship that the base cache remains more similar than the full KV cache across agents. The cache value gap is driven by adapter outputs, so the base cache consistently exhibits higher similarity than the full KV cache. Consequently, as agent heterogeneity increases and adapter outputs become more divergent, full KV cache similarity degrades more than base cache similarity. This means our method becomes *more* advantageous under greater heterogeneity, as illustrated in Figure 1 (left).
>
> Furthermore, our setup already reflects meaningful heterogeneity. Planning, action, and reflection are established as fundamentally distinct roles in prior multi-agent works (e.g., ReAct, Reflexion, ReWOO, BOLAA, AutoAct), inducing strong functional orthogonality. Due to space constraints, we omit the adapter weight similarity table, but we observed consistently low pairwise cosine similarity across agents, below 0.1, confirming adapter heterogeneity.
>
> In addition, we present additional experiments with a different role structure (debate) on a new dataset (MATH) following [1]. Table A consistently shows high base cache similarity and lower full-cache similarity, confirming the generality of our approach.
>
> Table A. Average cosine similarity between the 2 debate agents on the MATH dataset in LLaMA 3.1 8B Instruct.
>
> | Contributions | Full cache | Base cache |
> |---|---|---|
> | Cos Sim. | 0.9553 | 0.9766 |
>
> Regarding context overlap, Table B shows throughput gains scale approximately linearly with the overlap ratio, where BaseShared and BaseLRShared dominate next to FullShared.
>
> Table B. System throughput (tokens/sec) under context overlap ratios in 33.7K trajectory.
>
> | Overlap (%) | 100 (our paper) | 80 | 60 | 40 | 20 | 0 |
> |---|---|---|---|---|---|---|
> | Non-Shared | 683.2 | 683.2 | 683.2 | 683.2 | 683.2 | 683.2 |
> | FullShared | 1697.6 | 1385.1 | 1121.7 | 930.3 | 784.4 | 672.2 |
> | DroidSpeak | 931.0 | 859.0 | 789.3 | 742.4 | 703.8 | 677.9 |
> | BaseShared | 969.6 | 879.9 | 805.2 | 752.7 | 706.3 | 677.1 |
> | BaseLRShared | 1678.1 | 1375.6 | 1108.2 | 925.0 | 764.9 | 661.6 |
>
> >### **Response to Soundness W2, Presentation W2, and Question 1**
>
> We agree BaseLRShared may not directly apply to systems with already fixed independent down-projection matrices. However, when constructing a multi-LoRA agent system, the shared-A structure can be seamlessly adopted without changes to datasets or the LoRA formulation, and is likely to improve accuracy.
>
> The shared-A assumption covers a wide range of domains. Prior works (HydraLoRA and MTL-LoRA) show that sharing matrix A improves accuracy across diverse multi-task settings (e.g., BBH, GLUE, CommonsenseQA spanning 8+ domains), as matrix B alone suffices to capture heterogeneity while shared A enables more stable weights. We thus consider shared-A not as a limitation, but as an opportunity to build more accurate agent systems. It is expected to remain effective even as agent roles become more heterogeneous. In terms of efficiency, it introduces no computational overhead in new deployments since matrix A weight is simply identical without structural changes compared to conventional multi-LoRA.
>
> Furthermore, BaseShared alone already delivers substantial benefits without the shared-A assumption. It outperforms DroidSpeak in accuracy, reduces KV cache memory, and maintains competitive throughput ranking next to BaseLRShared and FullShared.
>
> >### **Response to Significance W1 and Question 2**
>
> When agents use different backbones or partially overlapping architectures, KV sharing becomes fundamentally limited. This is inherent to the problem and not specific to our method. KV sharing requires representational compatibility (same backbone) and contextual redundancy, and most prior agent frameworks and KV sharing studies similarly assume a homogeneous backbone. The shared-backbone assumption is thus a common constraint in this line of research.
>
> Importantly, even within the same-backbone setting, existing approaches fail to achieve efficient KV reuse. They are restricted to identical models without fine-tuning, or fail to leverage multi-LoRA structure despite high base cache similarity. Our work targets this practically important yet underexplored regime which we view as a prerequisite for tackling more heterogeneous settings. Regarding model scale, we plan to add results on a larger model as soon as possible.
>
> >### **Response to Presentation W1 and W3**
>
> We will revise the paper to explicitly state the setting to same-backbone multi-LoRA systems, and clearly separate claims for BaseShared (same-backbone) and BaseLRShared (shared-A).
>
>
> [1] https://llm-multiagent-ft.github.io

---

> > ### Author Rebuttal · Reviewer_4A6q · 2026-04-01
> >
> > Thank you for the detailed rebuttal and additional evidence. The response improves the clarity of the paper, especially by tightening the scope of the claims and adding overlap-ratio results. However, my main concerns are only partially addressed. In particular, the rebuttal still does not fully establish how the method behaves under increasing inter-agent heterogeneity, nor does it clearly characterize the applicability boundary of the shared-A assumption. These remaining concerns relate to the core scope and significance of the work and are not easily resolved within a short rebuttal. I will keep my score.

---

> > > ### Author Response · Authors · 2026-04-03
> > >
> > > We thank the reviewer for acknowledging and considering our previous response.
> > >
> > > We understand that the remaining concerns focus on (i) how our method behaves under increasing inter-agent heterogeneity, and (ii) the applicability boundary of the shared-A assumption. Due to space limitations in our previous response, we had to condense these points, so we would like to address them more in detail here.
> > >
> > > ---
> > >
> > > **First**, our results show consistent behavior across heterogeneous settings rather than relying on a limited or homogeneous setup. In our main configuration, the agent roles (planning, action, reflection) are known to exhibit strong functional differences, and we further confirm this at the parameter level by showing low pairwise cosine similarity (below 0.1) between LoRA adapter weights. In our additional experiment with a different role structure and task domain, namely debate-style agents on the MATH dataset (Table A), we again observe that base cache similarity remains high and consistently higher than full KV cache similarity.
> > >
> > > This clarifies a key point: our method does not depend on high homogeneity or high absolute base cache similarity. Instead, it relies on the relative relationship that base cache similarity is much higher than the full KV cache similarity, which holds across both setups. As heterogeneity increases, adapter-induced divergence mainly reduces full KV cache similarity, while the base cache remains comparatively stable. This widens the gap between the base and full cache similarity, under which full KV sharing methods (e.g., DroidSpeak) degrade more rapidly, whereas our base-cache-centered approach retains its advantage.
> > >
> > > We agree that the paper does not yet include accuracy evaluations under systematically controlled heterogeneity. This is mainly due to the limited diversity of established agent role configurations, as well as the limited availability of open-sourced multi-agent trajectory datasets. We will acknowledge this limitation in the revised paper and extend the evaluation when more suitable datasets become available.
> > >
> > >
> > > ---
> > >
> > > **Second**, regarding the applicability boundary of the shared-A assumption, our position is that shared-A remains effective under heterogeneous agent roles, as task-specific variation can be captured by the B matrices, while sharing A provides more stable and generalizable representations. This is supported by prior multi-task LoRA works (e.g., HydraLoRA and MTL-LoRA), which show strong performance across diverse domains such as the entire BBH, GLUE, and CommonsenseQA dataset. In our work, this is further supported by Table 7, where shared-A alone improves accuracy even without KV cache sharing. In addition, the mixed-dataset training result in Table D (response to reviewer k9PK) shows that combining datasets or domains does not degrade performance under shared-A.
> > >
> > > Importantly, shared-A is not required for our overall method. BaseShared applies broadly to same-backbone multi-LoRA systems without shared-A and already provides strong accuracy improvement and comparable efficiency over conventional KV cache sharing methods. BaseLRShared is an additional variant that provides further gains when shared-A is adopted.
> > >
> > > At the same time, we acknowledge that shared-A may become a bottleneck when the down-projection matrix A cannot be shared or jointly optimized. We will include these cases as failure modes and clarify this boundary in the revised manuscript.
> > >
> > >
> > > ---
> > >
> > > Again, we thank the reviewer for the detailed interest in our work. If any questions remain, please let us know.

---

### Decision · Program_Chairs · 2026-04-30

**Decision:**

Accept (regular)

**Comment:**

This paper considers an important problem in LLM inference efficiency: reducing the memory and compute overhead from redundant KV cache construction when multiple LoRA-adapted agents process shared trajectories in a multi-agent LLM system. LRAgent decouples the value cache into a shared base component and an adapter-dependent low-rank component, yielding BaseShared and BaseLRShared as two sharing schemes, paired with Flash-LoRA-Attention.

The rebuttal was substantive and all three reviewers independently recommended weak accept. Tables A, B, and C directly resolved the main empirical questions of Reviewer k9PK, who raised the score. Reviewer yxXo was satisfied and held a positive rating. Reviewer 4A6q acknowledged the improved clarity but maintained the score, citing remaining uncertainty about robustness under increasing role heterogeneity and the applicability boundary of shared-A.

Overall, the paper delivers a technically grounded contribution that fills a genuine gap in the literature on multi-LoRA agent systems. The Flash-LoRA-Attention kernel is a concrete artifact with clear reuse value. The primary limitations about evaluation scope and open questions around heterogeneity boundaries are real but do significantly not undermine the core contribution within the paper's stated setting. The rebuttal experiments (Tables A, B, C) should be incorporated into the final manuscript, and all presentation and framing issues listed above need to be corrected. With that, I recommend Accept.